# The involvement of the gut microbiota in postoperative cognitive dysfunction based on integrated metagenomic and metabolomics analysis

Shi-hua Zhang,[1,2] Xiao-yu Jia,[1] Qing Wu,[1,2] Jia Jin,[2] Long-sheng Xu,[1] Lei Yang,[1] Jun-gang Han,[1] Qing-he Zhou[1]

**ABSTRACT**  Cognitive dysfunction is a common symptom experienced by elderly individuals after surgery, resulting in memory problems, difficulties with logical thinking, hallucinations, delusions, and an increased risk of dementia. Despite its prevalence, the underlying cause of postoperative cognitive dysfunction remains unclear. Recent research has uncovered a link between neurodegenerative diseases and the gut microbiota, indicating the need for further investigation into the role of the intestinal flora in postoperative cognitive dysfunction. To address this research gap, we conducted behavioral tests, gene and protein analyses, metagenomics, and non-targeted metabolomics to compare the gut microbiota and metabolomics of mice exposed to anesthesia/surgery that exhibited cognitive impairment with those of age-matched control mice. Our goal was to identify possible correlations between these factors. We found that mice experiencing postoperative cognitive dysfunction had a distinct microbial composition, neuroinflammation, and synaptic damage compared to the control group. Specifically, we observed significant increases in the relative abundances of *Bacteroidetes unclassified*, *Bacteroides acidifaciens*, *Rikenellaceae bacterium*, *Muribaculaceae bacterium Isolate-104 HZI*, *Muribaculaceae bacterium Isolate-110 HZI*, and *Mucispirillum schaedleri* in aged mice exposed to anesthesia and surgery, while the relative abundances of *Lachnospiraceae bacterium A2*, *Lachnospiraceae bacterium A4*, *Lachnospiraceae bacterium*, *Blautia*, *Lachnoclostridium bacterium MD355*, *Eubacterium rectale*, *Ruminococcus sp. 1xD21-23*, and *Butyrivibrio* were significantly decreased. Additionally, metabolites, such as thiamine, spermidine, and long-chain unsaturated fatty acids, were down-regulated compared to the control group. These findings suggest that the intestinal metabolic abnormalities observed in elderly mice exposed to anesthesia/surgery may be regulated by the intestinal microbiota, specifically the *Lachnospiraceae*, *Lachnoclostridium*, *Butyrivibrio*, and *Eubacterium*. Therefore, our study highlights the potential of manipulating the gut microbiota to modulate the host metabolism in order to prevent and manage postoperative cognitive dysfunction.

**IMPORTANCE**  As the population ages and medical technology advances, anesthesia procedures for elderly patients are becoming more common, leading to an increased prevalence of postoperative cognitive dysfunction. However, the etiology and correlation between the gut microbiota and cognitive dysfunction are poorly understood, and research in this area is limited. In this study, mice with postoperative cognitive dysfunction were found to have reduced levels of fatty acid production and anti-inflammatory flora in the gut, and *Bacteroides* was associated with increased depression, leading to cognitive dysfunction and depression. Furthermore, more specific microbial species were identified in the disease model, suggesting that modulation of host metabolism through gut microbes may be a potential avenue for preventing postoperative cognitive dysfunction.

Address correspondence to Jun-gang Han, jghan@163.com, or Qing-he Zhou, zqh10980@zjxu.edu.cn.

Shi-hua Zhang and Xiao-yu Jia contributed equally to this article. Shihua Zhang is ranked first, since she wrote the manuscript.

The authors declare no conflict of interest.

**KEYWORDS** gut microbiota, metabonomics, anesthesia/surgery, postoperative cognitive dysfunction, dysbiosis

Cognitive dysfunction, characterized by cognitive impairment and attention deficits, is a common complication after surgery associated with several factors, including surgical trauma, advanced age, anesthesia, and pre-existing neurological conditions (1–3), among which anesthesia and surgery have been reported to be closely related to this condition (4). Clinical evidence indicates that postoperative cognitive dysfunction affects 10%–60% of elderly patients within the first week following surgery, with one-third experiencing long-term dysfunction (5, 6). Therefore, understanding the molecular mechanisms of postoperative cognitive dysfunction (POCD) is crucial for the search for potential therapeutic targets and clinical interventions.

The use of general anesthesia during early life has been hypothesized to be potentially associated with neuroinflammation and cognitive impairment, as evidenced by both preclinical studies and clinical trials (7–10). However, the precise etiology remains unclear and is likely to be multifactorial. Current research suggests that the pathogenesis may involve neuroinflammation, choline dysfunction, oxidative stress, and autophagy-related disorders (11, 12). Nonetheless, further investigation is necessary to determine the exact underlying mechanism. Therefore, a comprehensive and systematic exploration of the pathogenesis is crucial for establishing the relationships between the various existing hypotheses.

Research into the brain-gut axis, which investigates the interaction between the brain and the gut, has gained significance in recent years. Humans coexist in symbiosis with diverse microorganisms residing in the intestines, including bacteria, yeasts, archaea, viruses, protozoa, and even parasites such as worms, collectively referred to as the intestinal microbiota (13, 14). They have a reciprocal relationship with the host, with over $10^{14}$ bacterial cells distributed throughout the gastrointestinal tract (GI), with the vast majority ($10^{10}$–$10^{12}$ CFU/g of intestinal contents) located in the ileum and colon (15). The gut microbiota transforms various dietary components, such as macronutrients, micronutrients, fiber, and polyphenols, into a multitude of metabolites, including amino acid derivatives, vitamins, short-chain fatty acids (SCFAs), and trimethylamine (16). These metabolites and dietary components, derived from microbes, play critical roles in regulating the host's homeostasis, particularly with respect to the blood-brain barrier integrity and brain function (17–19). When the microbial balance is disturbed, the body's physiological metabolism undergoes corresponding changes, which are then communicated to the brain via circulatory or neural pathways. Therefore, there exists a co-metabolic process, whereby the body's metabolism is influenced by both its own functioning as well as the presence of the intestinal flora.

Accumulating evidence suggests a significant association between an imbalance in the intestinal flora and the onset of various neurological conditions, such as autism (20, 21), depression (22), schizophrenia (23), and Alzheimer's disease (AD) (24), which may potentially contribute to neurodegenerative diseases by facilitating the entry of molecules, including amyloid and lipopolysaccharides, across the blood-brain barrier. Microbiota in the gut have also been shown to influence the central nervous system (CNS) through the vagus nerve's release of cytokines, hormones, and immune signals (25–28), thereby highlighting an undeniable link between the gut-brain axis. In elderly mice, an imbalance of the intestinal flora is associated with decreased spatial cognition, learning, and memory (3, 29). Interestingly, moderate, low-intensity exercise and probiotic supplementation can attenuate neuroinflammation, alter the structure of the intestinal flora, and improve POCD in elderly mice (30). Several studies have described changes in the gut bacterial diversity in animal models of postoperative cognitive impairment using 16S rRNA sequencing technology. However, the specific species of microbes involved, their specific functions, and the extent to which these functions contribute to cognitive decline are still unknown. Additionally, the microbe-host co-metabolic relationship remains elusive. Further studies are therefore needed to

fully understand the specific microbial patterns associated with POCD and the potential role that co-metabolic processes play in cognitive decline.

To explore the potential involvement of gut microbes in cognitive impairment, we performed metagenomics and non-targeted metabolomics analyses using fecal samples. Aged mice exposed to anesthesia/surgery (AS) were initially examined for synaptic functions. Subsequently, using bioinformatics methods, we compared the microbial composition, function, and metabolites between the aged mice and those subjected to AS. Finally, network analysis was performed to integrate the multi-level omics data, providing valuable insights into how alterations in the gut microbiota and metabolites may impact behavior and enhance our understanding of the underlying mechanisms involved in these diseases.

## RESULTS

### AS-induced cognitive impairment and depressive behavior in aged mice

Morris Water Maze Test (MWMT) was conducted to evaluate the cognitive functions of elderly mice following AS. Prior to the experiment, all mice underwent 1 week of adaptation and behavioral training for 5 consecutive days. AS was performed on day 6, and behavioral tests and sampling were conducted 72 hours after the last administration of anesthesia (Fig. 1A). The results demonstrated that the aged mice receiving anesthesia exhibited significantly longer escape latency compared to the control group (Fig. 1C and D). Importantly, both groups of mice showed similar swimming speeds (Fig. 1F), indicating that postoperative movement and perception did not significantly affect spatial learning and memory. However, the multiple AS group displayed a reduced preference for the target quadrant in the platform-crossing experiment compared to the control group (Fig. 1B and E), suggesting an impairment in reference memory due to AS in aged mice. In the open field test (OFT), the multiple AS group exhibited a decreased central activity duration (Fig. 2A and C), reduced total distance traveled (Fig. 2B), a slower average speed (Fig. 2D), and an increased quiescent time (Fig. 2E) following exposure to isoflurane compared to the control group. These findings indicate that AS negatively impacts the locomotor activity of aged mice.

### AS caused synaptic damage in the hippocampus of aged mice, accompanied by neuroinflammation

Moreover, the synaptic changes in the hippocampus of aged mice following AS were investigated. Western blot analysis revealed a significant reduction in the postsynaptic membrane protein PSD95 in the hippocampus and the prefrontal cortex (Fig. 2F through H). Additionally, increased levels of interleukin-1β (IL-1β) and interleukin-6 (IL-6) were observed in the hippocampus (Fig. 2I), indicating neuroinflammation in response to anesthesia and surgery. These findings suggest that anesthesia and surgery may cause synaptic damage and neuroinflammation in the hippocampus and prefrontal cortex of aged mice, ultimately leading to cognitive impairment.

### AS-induced intestinal microbial metabolic disorders in aged mice

DNA was extracted from the distal colon contents of two groups of mice, and high-throughput sequencing using the NovaSeq 6,000 platform was conducted after quality control (QC) and data filtering. The diversity of the gut microbiota in elderly mice with AS-induced cognitive dysfunction was evaluated through α- and β-diversity analyses. The assessment of alpha-diversity involved calculating the Observed species, Chao1, Simpson, and Shannon indices. It was observed that both groups had similar alpha-diversity levels (Fig. 3A through D). However, a Principal Components Analysis (PCA) based on the weighted UniFrac distance matrix revealed significant differences in the intestinal microbial communities between the two groups (Fig. 3G). The results of Non-Metric Multidimensional Scaling analysis (stress = 0.03) and analysis of similarities (ANOSIM) ($R = 0.39$, $P = 0.010$) were consistent with the PCA results (Fig. 3H). Additionally,

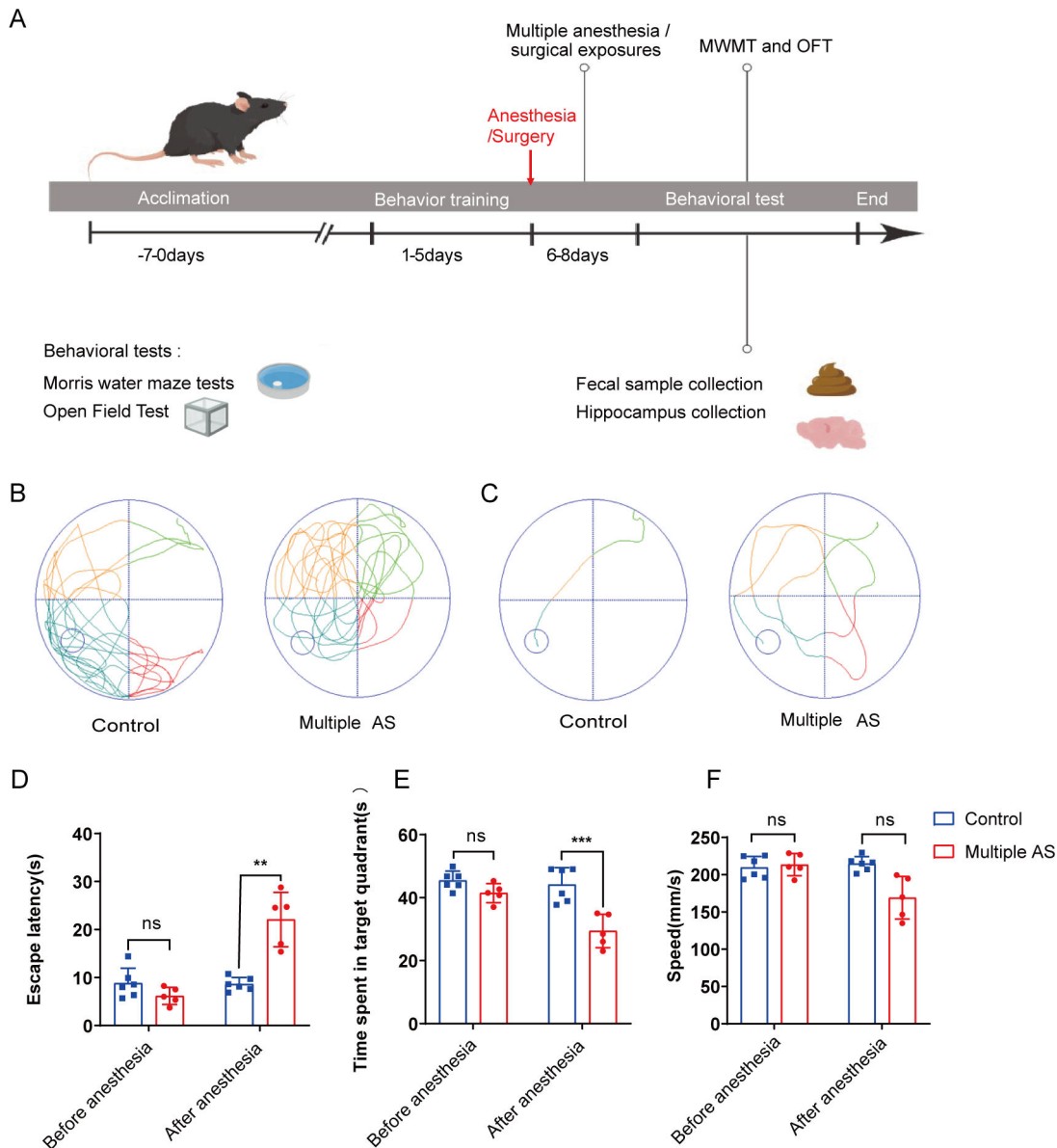

**FIG 1** AS exposure can impair spatial cognitive ability in aged mice. (A) Experimental timeline of behavioral tests (MWMT and OFT). (A–C) Behavioral trajectory of aged mice exposed to AS (B) in the probe test and (C) the hidden platform test. (D) The escape latency of reaching the hidden platform after AS for both the control and multiple AS groups. (E) The time spent in the target quadrant during the hidden platform test. (F) The swimming speed of each group was tested on the hidden platform. There was no statistically significant difference between the two groups. Results were displayed as mean ± SEM (control group: $n = 6$, multiple AS group: $n = 5$). Statistical significance is indicated with asterisks as follows: *, $P < 0.05$; **, $P < 0.01$; ***, $P < 0.001$; ****, $P < 0.0001$.

based on Core-Pan gene data, the sparse curve gradually flattened (Fig. 3E and F), indicating that the collected samples were suitable for subsequent bioinformatics analysis. Gene-level analyses were then conducted using non-redundant gene sets. The Upset analysis revealed 2722 unique species in the control group and 220 in the Multiple AS group, with 100,67 species shared between the two groups (Fig. 3I).

Furthermore, relative abundance analysis was performed to examine the phyla, genera, and species of bacteria. At the phylum level, the multiple AS group exhibited higher proportions of *Bacteroidetes*, *Proteobacteria*, *Verrucomicrobia*, and unnamed viruses, while *Firmicutes* and some unnamed bacteria had a lower proportion (Fig. 4A). The Mann-Whitney *U* test was employed to assess alterations in the intestinal flora at the genus and species levels within the two groups (Fig. 4B and C). AS-exposed mice showed

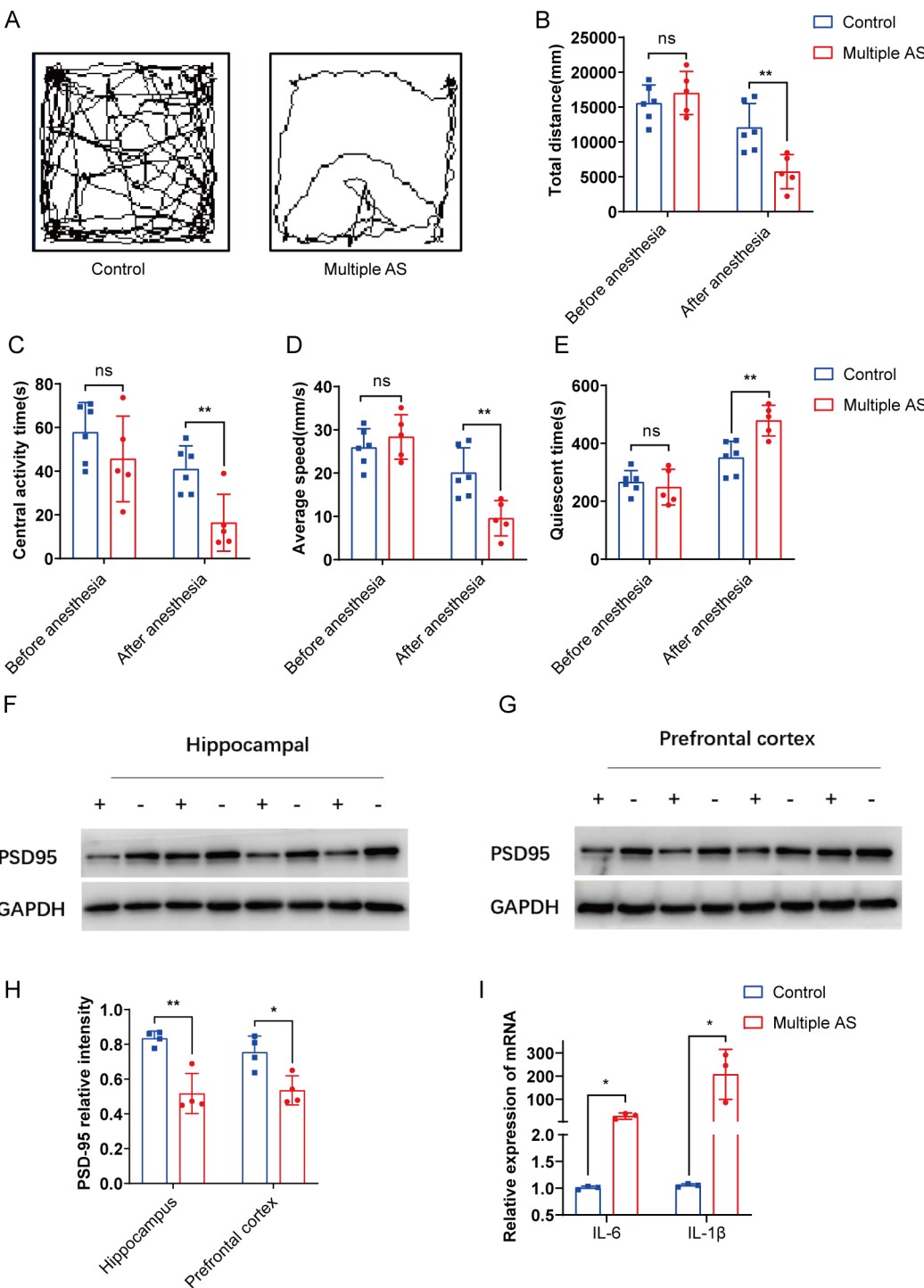

FIG 2 AS can not only lead to depression in aged mice but also cause synaptic plasticity damage and increased inflammation in the forebrain. (A) The route trajectory diagram in the OFT. (B–E) In the OFT experiment, the total distance of activity (B), the time of central activity (C), the average speed (D), and the rest time (E) of the mice were measured. (F) Western blot bands of PSD95 protein in the hippocampus. (G) Western blot band of PSD95 protein in the prefrontal cortex (+: multiple AS; −: control). (H) Relative quantification of the postsynaptic membrane protein PSD95 in the hippocampus and the prefrontal cortex. (I) The expression levels of IL-6 and IL-1β in the hippocampus (relative to glyceraldehyde-3-phosphate dehydrogenase (GAPDH) )were displayed as mean ± SEM (*n* = 4 per group). Statistical significance is indicated with asterisks as follows: *, *P* < 0.05; **, *P* < 0.01; ***, *P* < 0.001; ****, *P* < 0.0001.

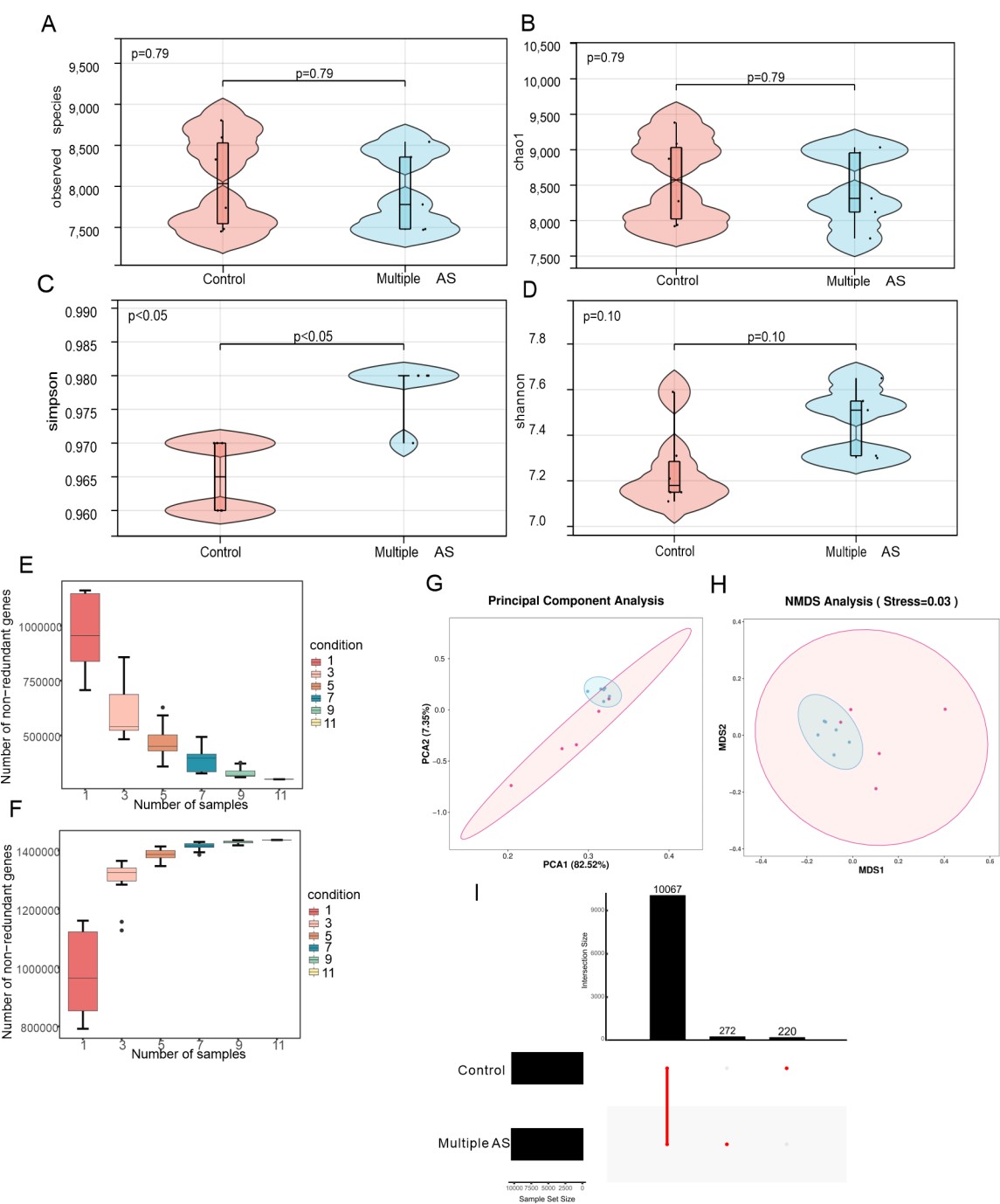

**FIG 3** Analysis of α diversity and β diversity between the control group and the multiple AS group. (A–D) Represent the alpha diversity indices of gut microbes between the control group and the multiple AS group (A: Observed Species, B: Chao1, C: Simpson, and D: Shannon). (E and F) The dilution curves of the Core(E) and Pan(F) genes. (G) The PCA plots were constructed using the Weighted UniFrac PCA method. (H) NMDS (Non-Metric Multidimensional Scaling) is a non-metric multidimensional scaling analysis. (I) UpSet Analysis Chart. Statistical significance is indicated with asterisks as follows: *, $P < 0.05$; **, $P < 0.01$; ***, $P < 0.001$; ****, $P < 0.0001$ (control group: $n = 6$ and multiple AS group: $n = 5$).

higher abundance of *s_Bacteroidales_unclassified*, *s_Lachnospiraceae_bacterium_28–4*, *s_Muribaculaceae_bacterium_Isolate-104_HZI*, *s_Muribaculaceae_bacterium_Iso-late-110_HZI*, *s_Bacteroides_acidifaciens*, *s_Mucispirillu_schaedleri*, and *s_Rikenella-ceae_bacterium* than the control mice. Conversely, *s_Ruminococcus_sp_1xD21-23*, *s_Eubacterium_plexicaudatum*, *s_Eubacterium_rectale*, *s_Lachnoclostridium_unclassified*, *s_Lachnospiraceae_bacterium_A2*, *s_Lachnospiraceae_bacterium_A4*, and

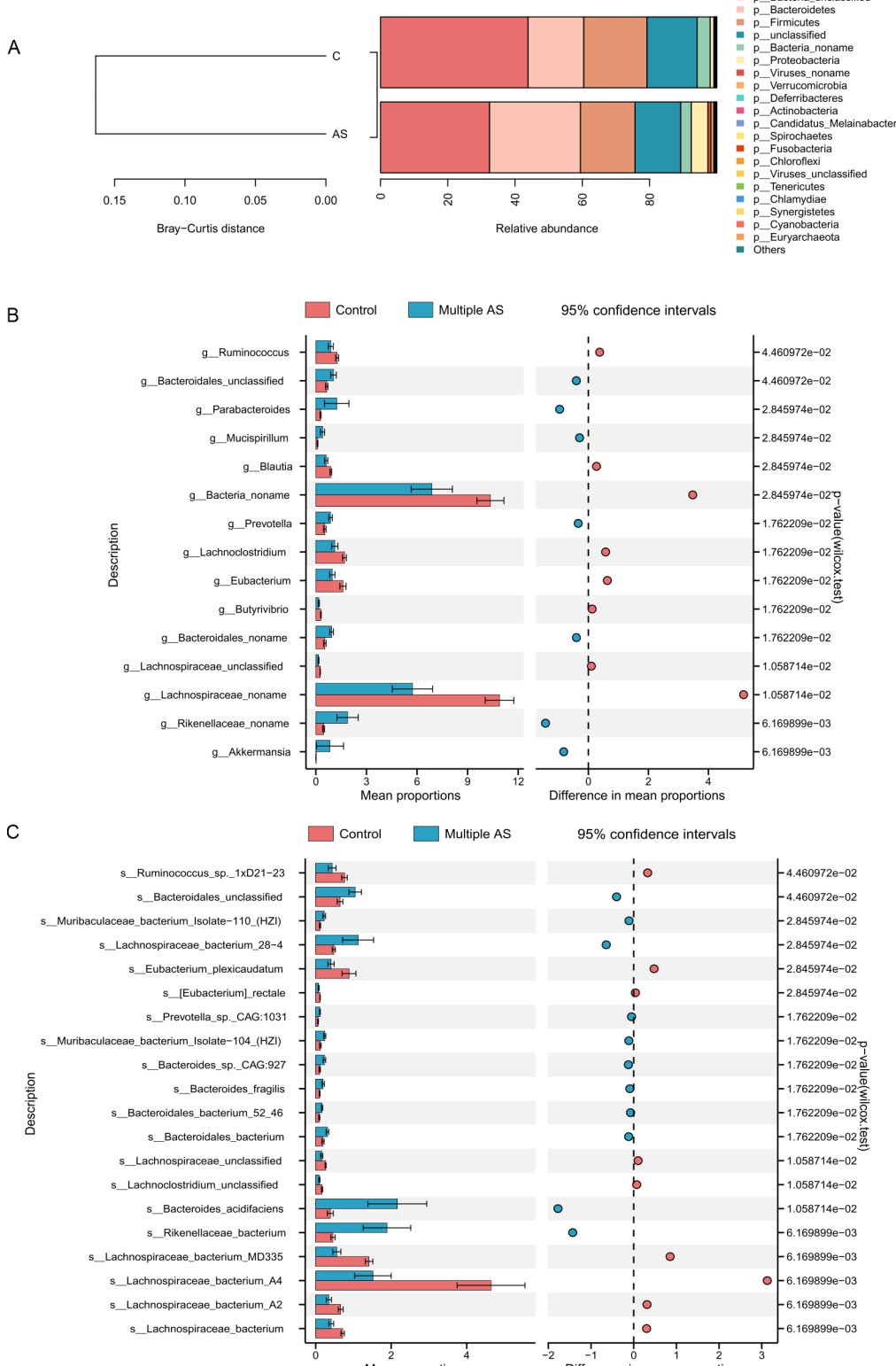

**FIG 4** Differences in intestinal microbes at the phylum, genus, and species levels between the control group and the multiple AS group. (A) Relative abundance at the phylum level. (B) The Stamp analysis revealed the relative abundance of the microbial flora at the genus level. (C) The relative abundance of different microbial populations obtained by Stamp analysis. Statistical significance is indicated with asterisks as follows: *, $P < 0.05$; **, $P < 0.01$; ***, $P < 0.001$; ****, $P < 0.0001$ (control group: $n = 6$ and multiple AS group: $n = 5$).

*s_Lachnospiraceae_bacterium* displayed an opposite trend. The box plots demonstrate comparable outcomes (Fig. S1).

Additionally, notable differences in microbial composition among the groups were detected using linear discriminant analysis effect size (LEfSe). This analysis revealed significant disparities in the hierarchical classification framework of the intestinal microflora, spanning from the phylum to the species level. In total, 53 different bacteria were identified across the phyla ($n = 5$), orders ($n = 6$), families ($n = 10$), genera ($n = 14$), and species ($n = 18$) levels (LDA >3, $P < 0.05$; Fig. 5A and B). Notably, the multiple AS group exhibited a higher abundance of *Bacteroidetes*, *Rikenellaceae*, *Akkermansiaceae*, *Verrucomicrobiae*, *Tannerellaceae*, *Prevotellaceae*, *Prevotella*, *Deferribacteraceae*, and *Mucispirillum*, while the control group was predominantly composed of *Lachnospiraceae* and *Bacteria*.

## Functional analysis of metagenomic sequencing revealed intestinal disorders in mice with AS-induced cognitive impairment

To compare the genes of functional bacteria, we focused on the results obtained from the KEGG database, which provides insights into the functional potential of the intestinal microbiota. Initially, a statistical evaluation was conducted to determine the disparity in

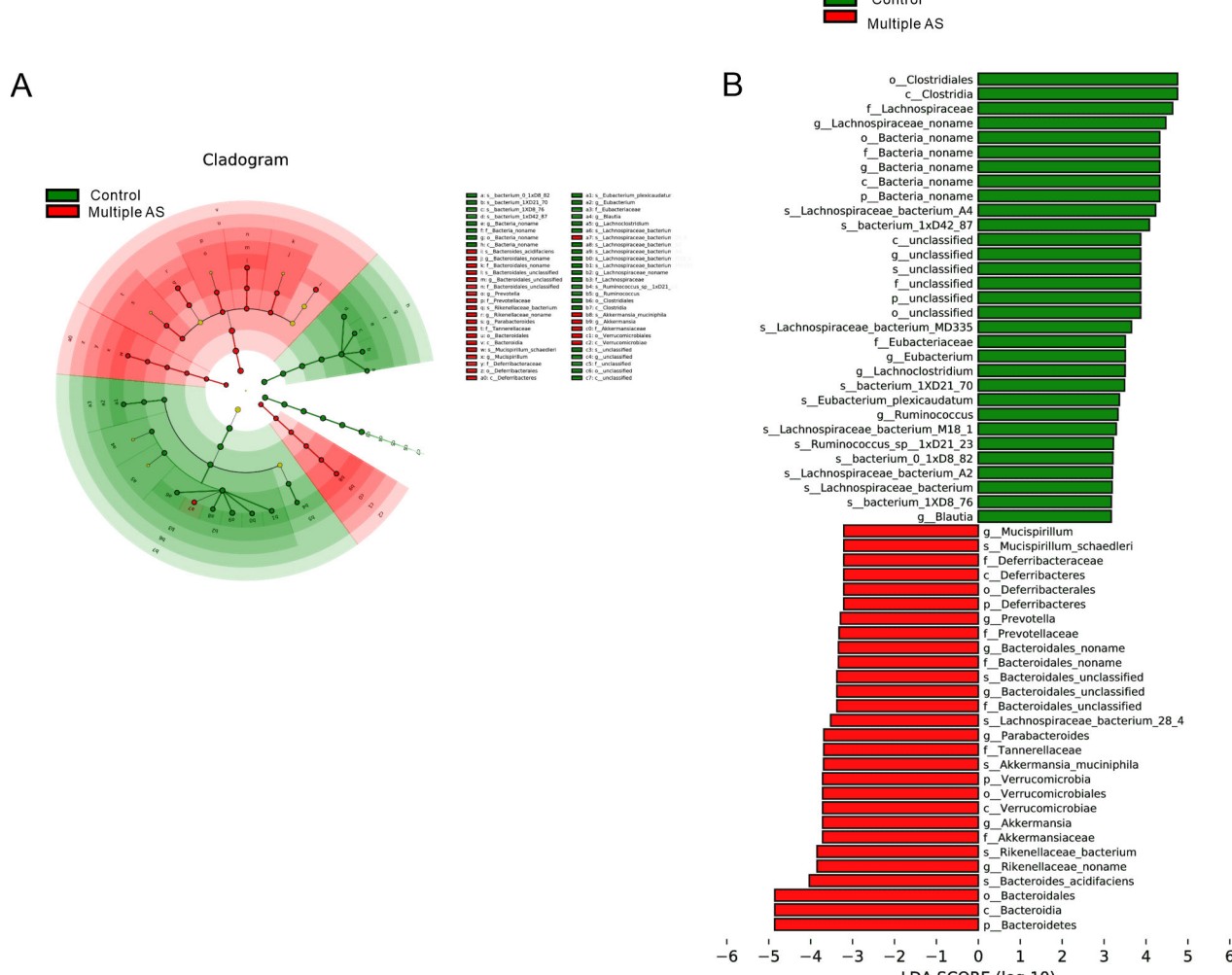

**FIG 5** LEfSe analysis was used to analyze the differences in the composition of the intestinal flora between the two groups. (A) Phylogenetic tree of the differential intestinal flora between the two groups. (B) The differential abundance microbial community classification branch diagram obtained by LEfSe analysis (LDA >3, $P < 0.05$; control group: $n = 6$; multiple AS group: $n = 5$).

the overall count of UniGenes in each group. The results showed that in the multiple AS group, there were 60,260 up-regulated UniGenes, while in the control group, 90,363 UniGenes exhibited up-regulation (Fig. 6A). Further analysis was performed at various levels of the KEGG classification, including KEGG Level 1, KEGG Level 2, KEGG Pathway Definition, and KEGG KO Description. At the highest level of classification, notable differences were observed in metabolic and genetic information-processing activities, which are known to be key functions of the gut microbiota in mice with AS-induced cognitive dysfunction (Fig. 6B). At the second level of classification, we identified

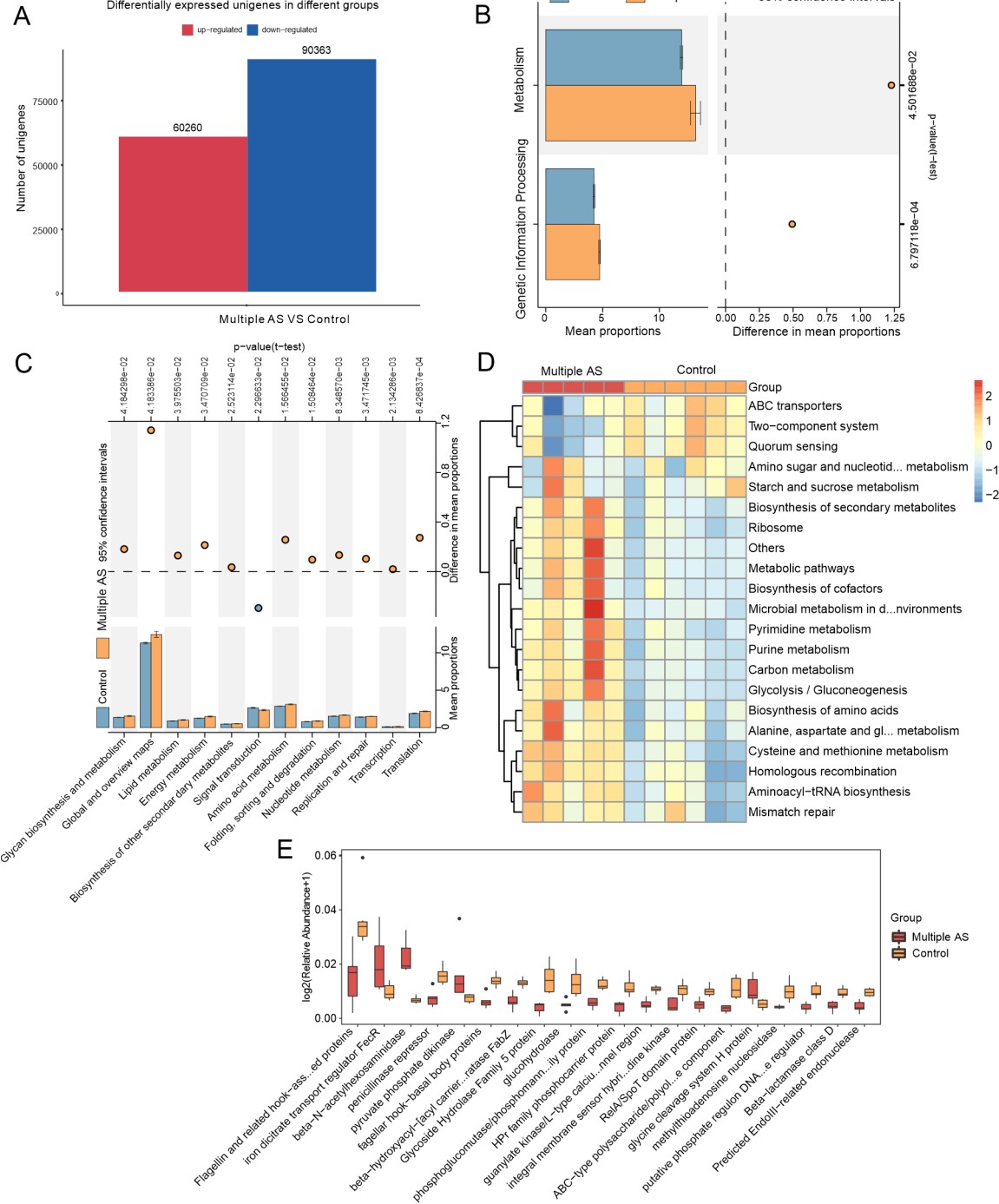

**FIG 6** Functional composition analysis of UniGene. (A) Differential expression analysis of Unigene. (B) An annotated analysis of KEGG at the first level. (C) KEGG Level2 analysis. (D) KEGG Pathway Definition Analysis. (E) KEGG KO Description Analysis.

alterations in nucleic acid metabolism, amino acid metabolism, glycine biosynthesis metabolism, translation, and energy metabolism in the gut microbiota of the multiple AS group compared to the control group (Fig. 6C). Furthermore, at the third level of classification, the gut microbiota of the Multiple AS group demonstrated an enrichment in biosynthesis pathways for secondary metabolites, purine metabolism, carbon metabolism, cysteine and methionine metabolism, the ribosome, and glycolysis/gluconeogenesis (Fig. 6D). These findings highlight significant disruptions in microbial metabolic function during AS-induced cognitive dysfunction in elderly mice. Detailed information of each pathway is shown in Fig. 6E.

## Cognitive impairment and depression after AS in elderly mice have been found to be associated with disorders in their intestinal microbial composition

A Spearman correlation analysis was conducted to investigate the association between an imbalance in gut microbiota and cognitive impairment as well as depression in elderly mice. We focused on comparing the differential flora at the genus and species levels with indicators of cognitive and depressive behaviors. The findings of the study demonstrated a positive correlation between *Akkermansia*, *Bacteroidales*, *Rikenellaceae*, *Mucispirillum*, *Parabacteroides*, and *Prevotella* with escape latency and quiescent time, while these microbial populations exhibited a negative correlation with the target quadrant time, center residence time, average velocity, and total distance (Fig. 7A). On the other hand, *Bacteria_noname*, *Lachnospiraceae_noname*, and *Lachnospiraceae_unclassified* were positively correlated with the target quadrant time, mean velocity, total distance, and central activity time but negatively correlated with escape latency and rest time. *Blautia* and *Eubacterium* were also negatively correlated with escape latency. In addition, *Lachnoclostridium*, *Ruminococcus*, and *Butyrivibrio* were positively correlated with the target quadrant time (Fig. 7A and B). At the taxonomic levels of genus and species, the aforementioned microbial populations demonstrated a consistent pattern, with *s_Muribaculaceae_bacterium_Isolate-110_(HZI)*, *s_Muribaculaceae_bacterium_Isolate-104_(HZI)*, *s_Bacteroides_sp_CAG:927*, *s_Bacteroides_fragilis*, and *s_Bacteroides_acidifaciens* all exhibiting a positive correlation with escape latency. These findings suggest the presence of a network association between the phenotype and gut microbiota at both the genus and species taxonomic ranks (Fig. 7C and D).

## AS-induced intestinal metabolite disorders in aged mice

Considerable evidence supports the continuous production of metabolites by gut microbes, which can be absorbed into the circulation and exert regulatory effects on the host's behavior and physiology. In order to explore the composition and biological functions of these metabolites, non-targeted metabolomics was performed using a high-resolution mass spectrometer. The total ion current diagram displayed the separation of all metabolites in liquid chromatography at each time point (Fig. S2A and C). The distribution of metabolite m/z-RT values and retention times (RTs) was visualized after peak alignment of mass spectrometry data using the XCMS software (Fig. S2B and D). Subsequently, we employed the open-source software metaX to annotate the m/z values of substances with the Kyoto Encyclopedia of Genes and Genome (KEGG) and Human Metabolome Database (HMDB) for primary metabolite identification. The results indicated a diverse range of identified metabolites, including benzene metabolites, lipids metabolism, organic acids metabolism, organic oxides, and organic heterocyclic compounds (Fig. 8A). Additionally, KEGG enrichment analysis was conducted on the annotated metabolites revealing 41 enriched KEGG pathways associated with various metabolic processes, such as amino acid metabolism, lipid metabolism, purine metabolism, bile acid metabolism, glycerophospholipid metabolism, and carbohydrate metabolism (Fig. 8B).

Comparative analysis of non-targeted metabolomics between the multiple AS group and the control group demonstrated significant differences in metabolite composition,

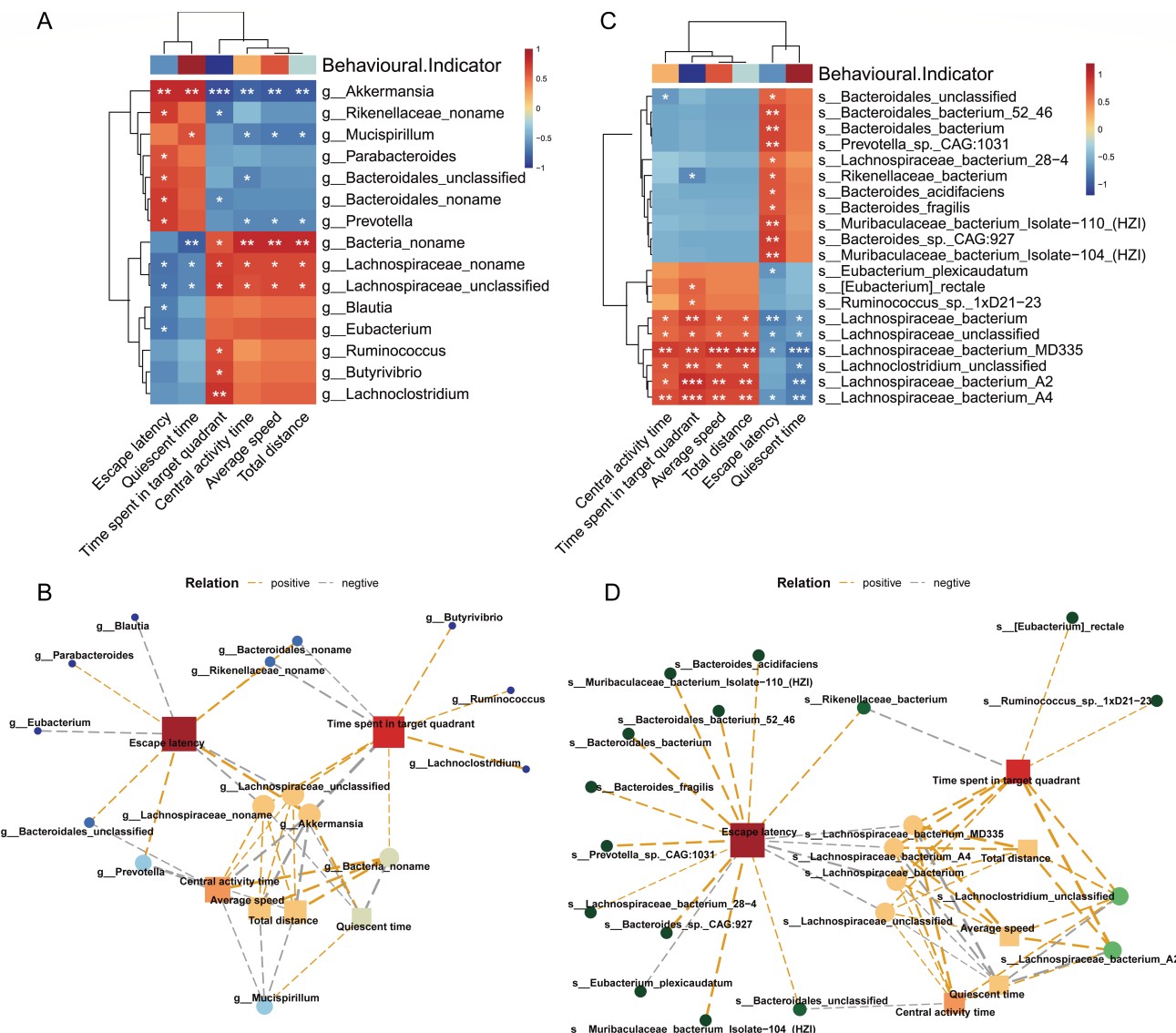

**FIG 7** Cognitive impairment and depression in aged mice after AS exposure is related to an imbalance in their intestinal flora. (A) Correlation between the differential flora at the genus level and the behavioral phenotype after AS exposure. (B) This is a diagram depicting the links between microbiota and behavior in elderly mice after they have undergone AS. (C) The correlation between different flora and behavioral phenotypic changes at the species level. (D) This is a map of the network interactions of different flora and behavioral phenotypic changes at the species level. Statistical significance is indicated with asterisks as follows: *, $P < 0.05$; **, $P < 0.01$; ***, $P < 0.001$; ****, $P < 0.0001$.

observed in both positive and negative ion modes (Fig. 9C). Partial Least Squares Discriminant Analysis (PLS-DA) showed a distinct separation in the overall metabolomic characteristics between elderly mice in the multiple AS group and the control group. The validity of the PLS-DA model was confirmed by a permutation test, indicating non-over-fitting with R2Y and Q2 values of 0.93 and −0.83, respectively (Fig. 9A and B). In the positive ion mode, 1,722 features were up-regulated, and 1,001 features were down-regulated, while in the negative ion mode, 430 features were up-regulated, and 294 features were down-regulated. Overall, 2,723 features were identified in the positive ion mode and 724 features in the negative ion mode (Fig. 9C; Fig. S3). These findings suggest disruption of the intestinal metabolites in mice with AS-induced cognitive dysfunction. Moreover, a comprehensive analysis revealed the identification of 112 distinct metabolites in fecal samples, differentiating the control and multiple AS groups. Notably, a majority of these metabolites displayed decreased expression, highlighting significant

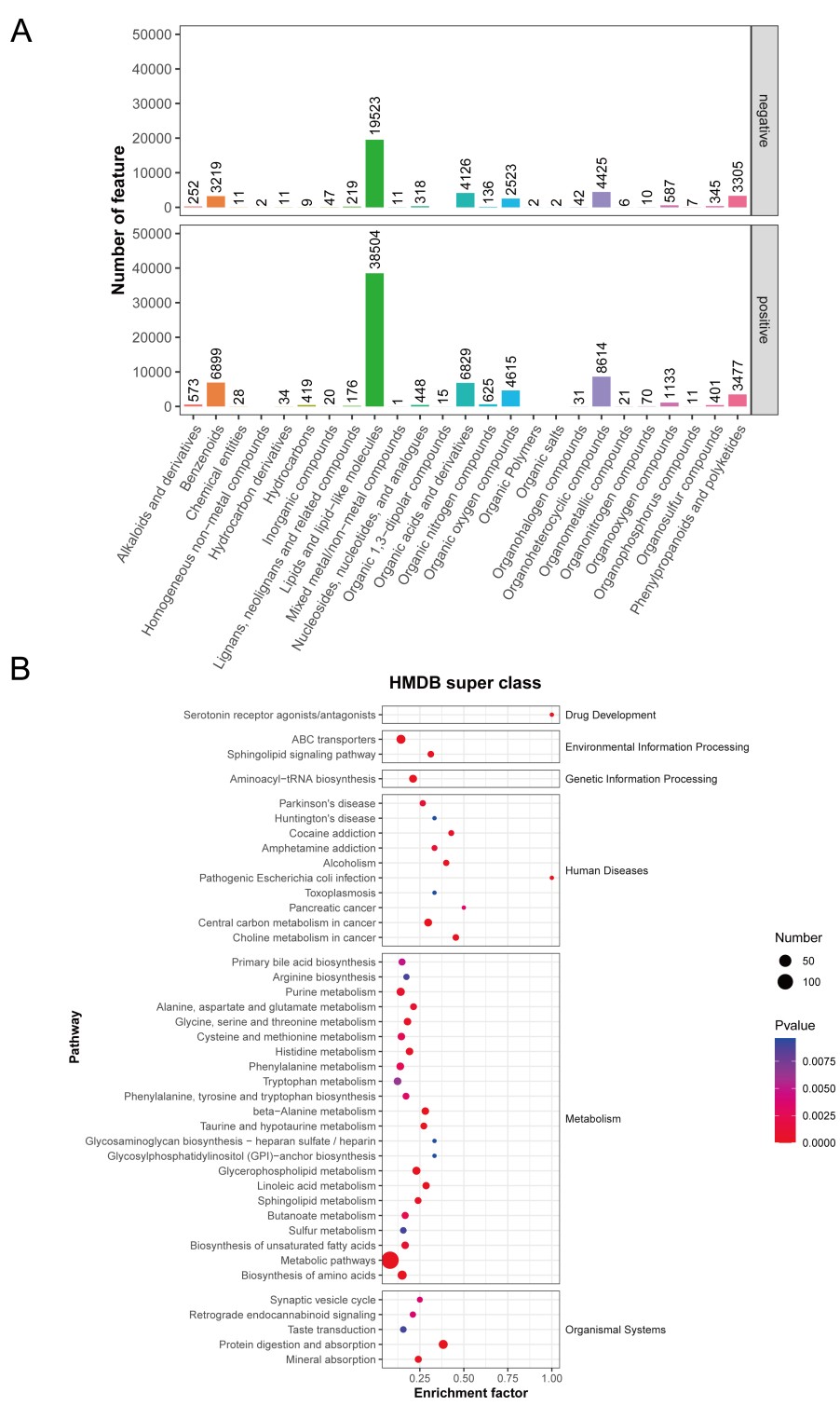

**FIG 8** Metabolite identification results map. (A) HMDB Super class classification diagram. The Super-class entry is in the horizontal coordinate, corresponding to the ordinate of the number of metabolites in the entry. (B) Enrichment analysis of secondary metabolites in KEGG pathways.

alterations in their levels (Fig. 9D). The hierarchical heatmap provided visual evidence of the changing pattern of intestinal metabolites, confirming substantial distinctions between the multiple AS group and the normal group (Fig. 9D). KEGG enrichment analysis further revealed that these differential metabolites were enriched in purine

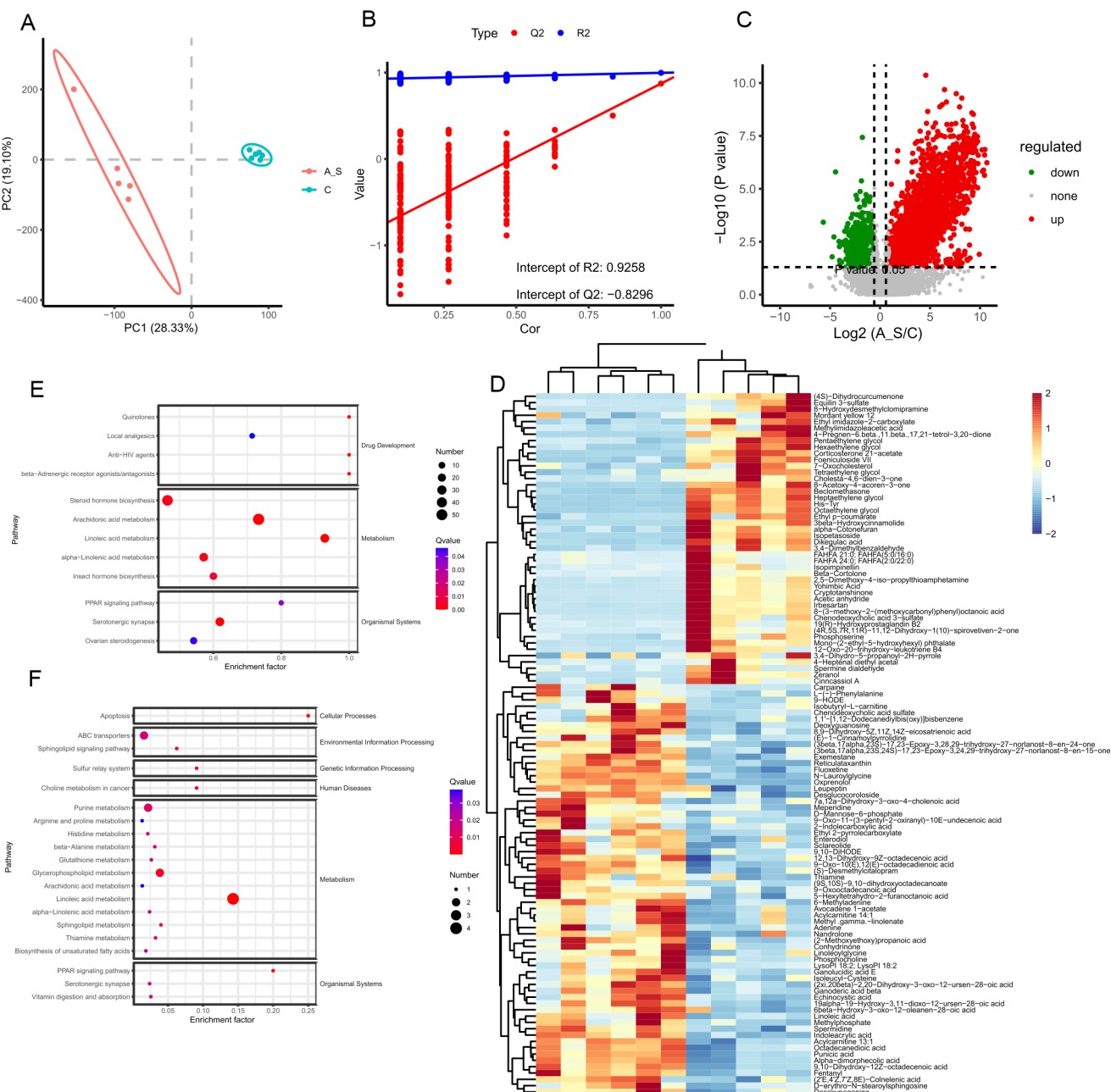

**FIG 9** Anesthesia/surgical exposures broke the balance of intestinal metabolism in aged mice. (A) The PLS-DA score plot: the abscissa represents the first principal component (PC1), and the ordinate represents the second principal component (PC2). (B) The permutation test diagram. R2 represents the interpretation rate of the model on the Y-axis direction, and Q2 represents the prediction rate of the model. (C) According to the conditions of variable importance in projection (VIP) > 1 and $P < 0.05$, the confidence interval of the volcano plot of the metabolites separated by the model is 95%. Red dots represent significantly up-regulated metabolites, green dots represent significantly down-regulated metabolites, and gray dots represent metabolites with no significant changes. (D) Thermograms of differential metabolites. (E–F) The bubble map of KEGG enrichment analysis of differentially metabolized compounds is shown.

metabolism, amino acid metabolism, linoleic acid metabolism, choline metabolism, apoptosis, and glycerophospholipid metabolism (Fig. 9E and F).

## AS-induced memory impairment and depression in aged mice are related to microbial-host metabolic interactions

To further investigate the relationship between alterations in intestinal metabolite abundance and intestinal flora, a selection of 35 metabolites with KEGG annotation was made based on the criteria of variable VIP > 1, ratio ≥1.5 or ≤1/1.5, and $P$-value ≤ 0.05 (Fig. 10A). Spearman correlation analysis was performed to assess the covariant relationship between 20 species and the selected metabolites, resulting in the generation of a relevant network to illustrate the connections (Fig. 10D). The *s_Lachnospiraceae_unclassified*, *s_Lachnospiraceae_bacterium_MD335*, *s_Lachnospiraceae_bacterium_A4*, *s_Lachnospiraceae_bacterium_A2*, *s_Lachnoclostridium_unclassified*, and *s_Ruminococcus_sp_1xD21-23* were found to be positively correlated with thiamine, spermidine, phosphocholine, acylcarnitine, linoleic acid, D-erythro-N-stearoylsphingo-sine, α-dimorphecolic acid, 9,10-dihydroxy-12Z-octadecenoic acid, 8,9-dihydroxy-5Z, 11Z, 14Z-eicosatrienoic acid, 7α, 12α-dihydroxy-3-oxo-4-cholenoic acid, 6-methyladenine, and (9S, 10S)−9,10-dihydroxy-12Z-octadecanoic acid [long-chain unsaturated fatty acids (LCPUFAs) Fig. 10C and D]. In addition, at the genus level, *Butyrivibrio*, *Lachnospiraceae*,

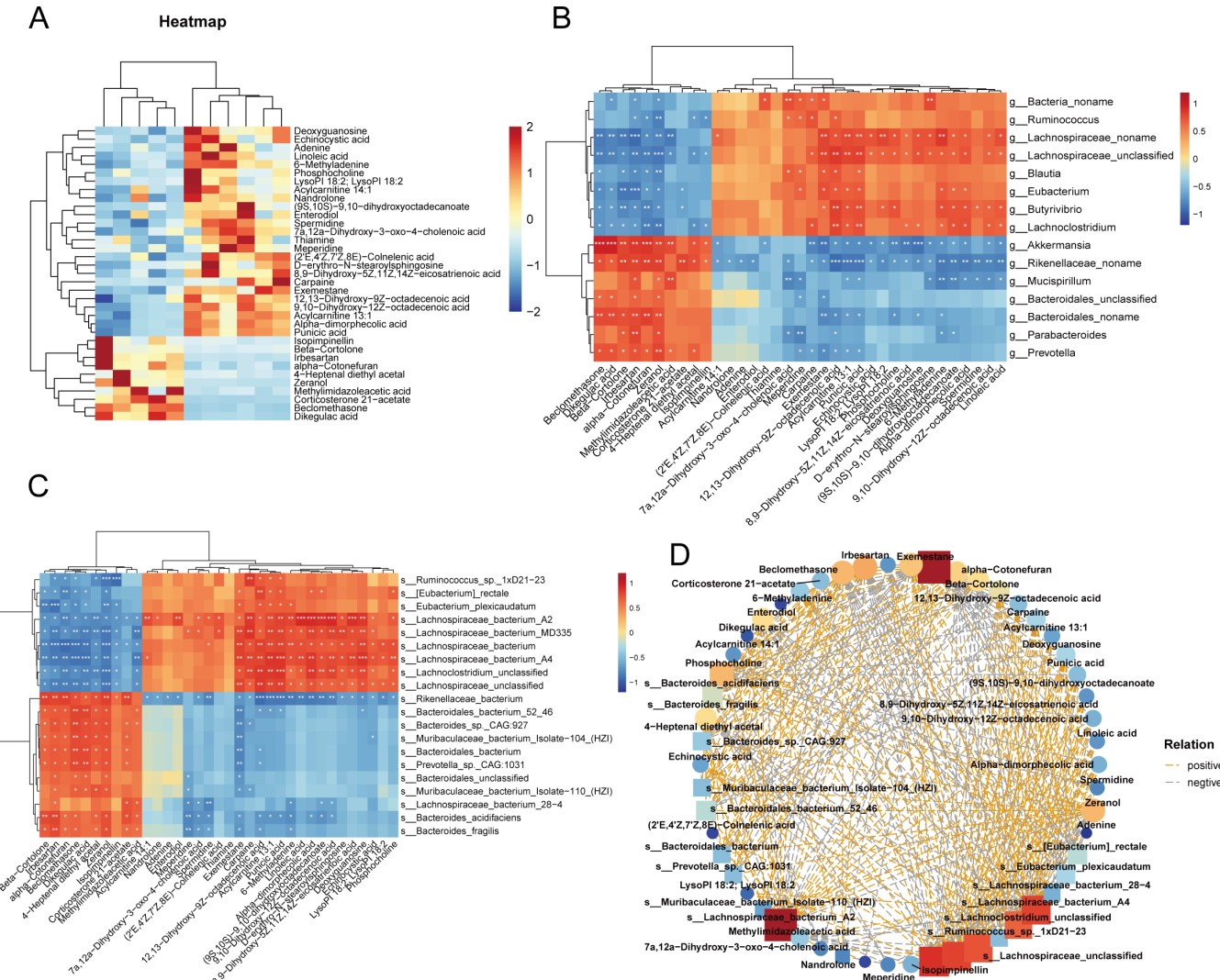

**FIG 10** Analysis of multi-omics network regulation in the AS exposure group and the blank control group. (A) Differential metabolite heat maps based on VIP > 1, $P < 0.05$, and KEGG annotation screening. (B and C) Heatmap of correlation analysis of differential microbial-host interactions. (D) Correlation network of differential metabolites and microbiota between the control and multiple AS groups. Statistical significance is indicated with asterisks as follows: *, $P < 0.05$; **, $P < 0.01$; ***, $P < 0.001$; ****, $P < 0.0001$ (control group: $n = 6$ and multiple AS group: $n = 5$).

and *Lachnociostridium* showed positive correlations with spermidine, phosphocholine, acylcarnitines, LCPUFAs, and thiamine (Fig. 10B). Conversely, the *Rikenellaceae*, *Akkermansiaceae*, and *Mucispirillum* exhibited an opposite trend (Fig. 10B).

## DISCUSSION

Aging leads to a deterioration of cellular, tissue, and organ functions, accompanied by susceptibility to chronic inflammation in the CNS and GI system, which are particularly sensitive to metabolic dysregulation. With aging, there is an increase in the number of goblet cells, while the expression of alpha-defensins, lysozymes, and F4/80 mRNA, along with NOx levels and protein concentrations of tight junction proteins, decreases (31, 32). These changes are associated with elevated intestinal permeability and increased levels of bacterial endotoxins (33). Several studies have linked these age-related modifications to alterations in the gut microbiome observed in elderly humans and animals (34–36). In a recent study, it was found that transplanting fecal matter from aged individuals into young mice leads to disruption of the intestinal epithelial barrier and heightened levels of inflammation, particularly in the retina and nervous system. Neuroinflammation in older mice with young donor microbiota was reversed by enriching B vitamins and lipid synthesis pathways, implying an important role in gut microbial metabolism in aging (37). Additionally, POCD may possess multifactorial origins, such as an intraoperative inflammatory response, anesthesia, and aging, further increasing susceptibility. In this study, we aim to investigate the contribution of gut microbes to POCD in elderly mice.

Cognitive dysfunction usually occurs after surgery in elderly patients, characterized by impaired memory, logical thinking, hallucinations, and delusions (38). In this study, we examined the behavior, intestinal flora, metabolites, IL-6 and IL-1β hippocampal levels, and synaptic function of elderly mice with AS-induced cognitive dysfunction. Our findings revealed significant behavioral alterations, neuroinflammation, synaptic dysfunction, and disruptions in the intestinal flora and metabolites in the aged mice compared to normal mice. Specifically, older mice exhibited a longer escape latency and a shorter time spent in the target quadrant in the MWMT. This finding is consistent with previous studies (39). Moreover, synaptic damage was observed in the hippocampus and the prefrontal cortex, further supporting the presence of cognitive impairments. Elevated levels of IL-6 and IL-1β were also detected in the hippocampus, suggesting the occurrence of neuroinflammation. Neuroinflammation has been associated with memory impairment, brain injury, depression, and other neuropsychiatric disorders (40–45). Studies have proposed that changes in the intestinal flora can induce neuroinflammation, leading to the release of pro-inflammatory cytokines including CRP, IL-1β, IL-6, and TNF-α, and the disruption of the blood-brain barrier, which may activate or impair astrocytes and contribute to neurodegenerative disorders (46, 47). These findings suggest a potential link between surgery/anesthesia-induced POCD and alterations in the gut flora. However, further research is necessary to fully elucidate the underlying mechanisms.

Interestingly, our study uncovered a new observation: that aged mice that have undergone surgery/anesthesia exhibit both reduced cognitive memory and depressive characteristics. The Open Field Test revealed decreased autonomous activity and exploration abilities, accompanied by signs of depression. Previous studies have reported that AS can induce anxiety behavior by affecting the intestinal microbiota in young rats (48), but no depression tendency was observed. Investigations into the relationship between intestinal microbial diversity and depression have shown that specific bacteria, including *Bacteroides*, *Proteus*, and *Actinomycetes*, exhibit a positive correlation with the presence of depression (49, 50), while *Firmicutes* display a negative correlation with the occurrence of depression (51). Depression can be influenced by alterations in these bacterial groups' proportions of these bacterial groups. Our findings align with these studies, as we observed an increased proportion of *Bacteroidetes* and *Proteobacteria* and a *decrease* in the proportion of *Firmicutes*. Additionally, a recent study found that repeated exposure to sevoflurane in early life altered emotional responses in primates,

which supports our findings (52). Taken together, our results suggest that AS can induce depression in aged mice, likely through alterations in the gut microbiota.

Previous studies on gut microbiota composition in mice with cognitive dysfunction have predominantly utilized 16S rRNA analysis, which is limited to identifying microorganisms at the genus or family level. In our study, we employed metagenomic sequencing to investigate the intestinal microflora in greater detail, identifying microorganisms at the species level. Our findings revealed that, compared to the control group, the relative abundance of specific species, such as s_Bacteroidales_unclassified, s_Muribaculaceae_bacterium_Isolate-104_HZI, s_Muribaculaceae_bacterium_Isolate-110_HZI, s_Bacteroides_acidifaciens, s_Mucispirillu_schaedleri, and s_Rikenellaceae_bacterium, was increased in mice exposed to AS. Conversely, species, such as s_plexicaudatum, s_[Eubacterium]_rectale, s_Lachnoclostridium_unclassified, s_Lachnospiraceae_bacterium_A2, s_Lachnospiraceae_bacterium_A4, and s_Lachnospiraceae_bacterium, showed an opposite trend. Bacteroidales is a common flora in the human intestine, and studies have shown that it has a high proportion in patients with severe depression (51, 53–55). Our correlation analysis also revealed that both the genus and species levels of Bacteroidales are negatively correlated with the central activity time, which is consistent with our findings. Furthermore, a separate investigation examining the impact of AS-induced POCD observed a decline in the abundance of Bacteroidales_unclassified (56), implying that varying modes of exposure could yield divergent outcomes. In a study assessing the effects of baicalein treatment on memory and cognitive impairments in APP/PS1 mice, s_Muribaculaceae_bacterium_Isolate-104_HZI, and s_Muribaculaceae_bacterium_Isolate-110_HZI were identified as potential biomarkers (57). Our results showed an increasing expression trend in both of these bacteria, which led us to hypothesize that the cognitive impairment caused by AS might cause a stress response in the early stages to defend against external stimulation. These two bacteria were positively correlated with escape latency, suggesting that their increased abundance can lead to longer escape times. Furthermore, s_Mucispirillum_schaedler has been linked to cognitive impairment caused by Porphyromonas gingivalis, indicating its potential significance as a marker of cognitive impairment (58). It is necessary to do further research to verify this. The relative abundance of some beneficial bacteria such as s_Eubacterium_plexicaudatum (59) and s_Eubacterium_rectale (60, 61), which have anti-inflammatory effects, decreased in mice following AS. The relative abundance of Lachnospiraceae (62), Blautia, Eubacterium (63), and some Lachnoclostridium (64) producing SCFAs also decreased. SCFAs possess the ability to traverse the blood-brain barrier, thereby exerting regulatory effects on brain function. This regulation can be achieved through two mechanisms: the enhancement of the intestinal barrier and the inhibition of bacterial and bacterial product translocation, or direct interaction between SCFAs and immune cells. Consequently, the indirect reduction of systemic inflammation facilitated by SCFAs can subsequently mitigate neuroinflammation within the brain. Studies in rodents have indicated that SCFA levels and brain function can be altered with the administration of prebiotics and probiotics (28). For instance, in the context of a vascular dementia model, the administration of Clostridium butyricum to mice resulted in elevated levels of butyrate in both feces and brains, causing cognitive impairment and histopathological alterations in the hippocampus, along with a reduction in the expression of proteins associated with the BDNF-PI3K-AKT pathway in the hippocampus (65, 66). Our correlation analysis results suggest that a decrease in these beneficial bacteria is linked to cognitive impairment and depression, as it was associated with longer target quadrant times in the MWMT, shorter escape latencies, faster average speeds in the OFT, longer central activity times, and stronger activity abilities.

Each individual's metabolic characteristics are unique, and changes in metabolite concentrations provide insight into underlying physiological mechanisms and the condition of the disease. Nucleotides, fatty acids, amino acids, and neurotransmitters are usually involved in metabolic disorders caused by AS (56). Metabolomics analysis of AD has also revealed significant disruptions in various metabolic pathways and reactions,

such as lipid homeostasis, fatty acid biosynthesis, amino acid metabolism, mitochondrial bioenergetics, reactive oxygen species neurotransmitter biosynthesis, synaptic transmission, calcium homeostasis, inflammation/immune response, and apoptosis (67, 68). In line with these findings, our metabolomics analysis of aged mice exposed to AS showed distinct metabolic abnormalities compared to the control group. These differentially regulated metabolites were primarily enriched in purine metabolism, amino acid metabolism, linoleic acid metabolism, glycerophospholipid metabolism, choline metabolism, and apoptosis. Notably, we observed the down-regulation of synthetic neurotransmitter precursors, such as choline, spermidine, and thiamine, in mice with AS-induced cognitive impairment. These substances play crucial roles not only in the synthesis of neurotransmitters, such as acetylcholine and GABA, but also in the synthesis of membranes and the maintenance of synaptic functions (67–70). In AD patients, the phosphorylation-dephosphorylation process of thiamine is disrupted, and the activities of phosphatases (TMPase and TDPase) are decreased (71). Recent studies have indicated that exogenous supplementation of thiamine (72), choline (73), and spermidine (74–78) can enhance cognitive ability and improve mitochondrial function. These findings suggest a potential link between POCD induced by AS and the homeostasis of these substances. Additionally, we observed down-regulation of LCPUFAs and their derivatives, including (2'E, 4'Z, 7'Z, and 8E)-Colnelenic acid, Linoleic acid, 9,10-Dihydroxy-12Z-octadecenoic acid, 12,13-Dihydroxy-9Z-octadecenoic acid, 8,9-Dihydroxy-5Z, 11Z-14Z-eicosatrienoic acid, Punicic acid, and (9S, 10S)−9,10-dihydroxyoctadecanoate. A multitude of clinical and animal studies have consistently emphasized the significance of LCPUFAs in the processes of neurodevelopment and neurodegeneration (79). Conditions such as schizophrenia and attention deficit hyperactivity disorder have been linked to inadequate levels of LCPUFAs (80–82). They play a critical role in maintaining the integrity and function of neuronal membranes and are essential nutrients for brain and visual system development (83), and an adequate supply of LCPUFAs is necessary for neural development (84). Considering the observed synaptic damage and neuroinflammation, we hypothesize that reductions in metabolites, such as choline, thiamine, spermidine, and LCPUFAs, may contribute to cognitive impairment and depression following anesthesia/surgical exposure. Metabolomics offers insights into the metabolic changes that may be the underlying cause or consequence of cognitive and emotional issues following anesthesia or surgical exposure. Taken together, our exploratory metabolomics study of fecal samples from aged mice with AS-induced cognitive impairment revealed disturbances in fatty acid metabolism, lipid metabolism, and thiamine metabolism, among others. However, these findings are not limited to just these pathways. According to our multi-omics analysis, these cognitive impairments may be caused by metabolic disorders in the intestinal microbiota. Further experimental verification is currently underway to support these findings.

To gain insight into the role of gut microbiota in the metabolism of POCD, this study analyzed the relationship between metagenomics and metabolomics. Results demonstrated that beneficial bacteria, including s_Lachnospiraceae_bacterium_A2, s_Lachnospiraceae_bacterium, s_Lachnoclostridium_unclassified, and s_Eubacterium_rectale were positively correlated with some metabolites such as choline phosphate, LCPUFAs, and spermidine. Butyrivibrio and Lachnospiraceae were all positively correlated with spermidine, phosphocholine, and LCPUFAs at the genus level. It has been reported that many microorganisms in the intestine can synthesize LCPUFAs (85, 86) and spermidine (87). In addition, it has been shown that the gut microbiota can produce spermidine through compositional or inducible amino acid decarboxylases (87, 88). The synthesis of spermidine involves two enzymes: S-adenosylmethionine decarboxylase (AdoMetDC) and the aminopropyl transferase spermidine synthase (SpdSyn). These enzymes are present in a wide range of organisms, including nearly all eukaryotes, most archaea, and various bacterial phyla such as Escherichia coli, Bacillus subtilis, and Thermotoga maritima. Surprisingly, many bacterial species that do not contain AdoMetDC or SpdSyn can also synthesize spermidine, such as Proteobacteria, Clostridium, Eubacterium,

*Ruminococcus*, and *Butyvibrio* (87). Our study revealed that *Butyrivibrio*, *Lachnociostridium*, and *Lachnospiraceae* were down-regulated, which was consistent with the trend of LCPUFAs and spermidine. This finding suggests that the reduction in these metabolites may be attributed to the down-regulation of these specific microbial flora. Therefore, it is hypothesized that AS might lead to a decrease in the gut microbiota involved in producing LCPUFAs, spermidine, and choline, resulting in reduced levels of these substances and other components involved in synthesizing intestinal neurotransmitters. This information can be transmitted directly to the brain via the vagus nerve or the gut autonomic nerve, triggered by local gut microbial metabolites (Fig. 11). Additionally, the disturbance of the intestinal microbial flora has the potential to initiate the secretion of peripheral pro-inflammatory cytokines, namely IL-1β and IL-6, resulting in heightened permeability of the blood-brain barrier, disruption of the CNS, and subsequent cognitive impairment. Consequently, further investigations are required to identify precise microorganisms linked to neural pathways regulated by metabolites such as LCPUFAs, choline, and spermidine. On the other hand, the *Rikenellaceae*, *Akkermansiaceae*, and *Mucispirillum* exhibited an opposite trend, suggesting that the decrease in long-chain fatty acids, spermidine, and choline may be regulated by these particular microbial florae, although further experimental verification is required. There are some limitations to this study. On the one hand, small sample sizes may make some differences appear larger. On the other hand, there are certain differences between animal models and human diseases that do not fully simulate clinical outcomes.

Overall, the analysis of the relationship between metagenomics and metabolomics provides valuable insights into the potential molecular mechanisms underlying POCD

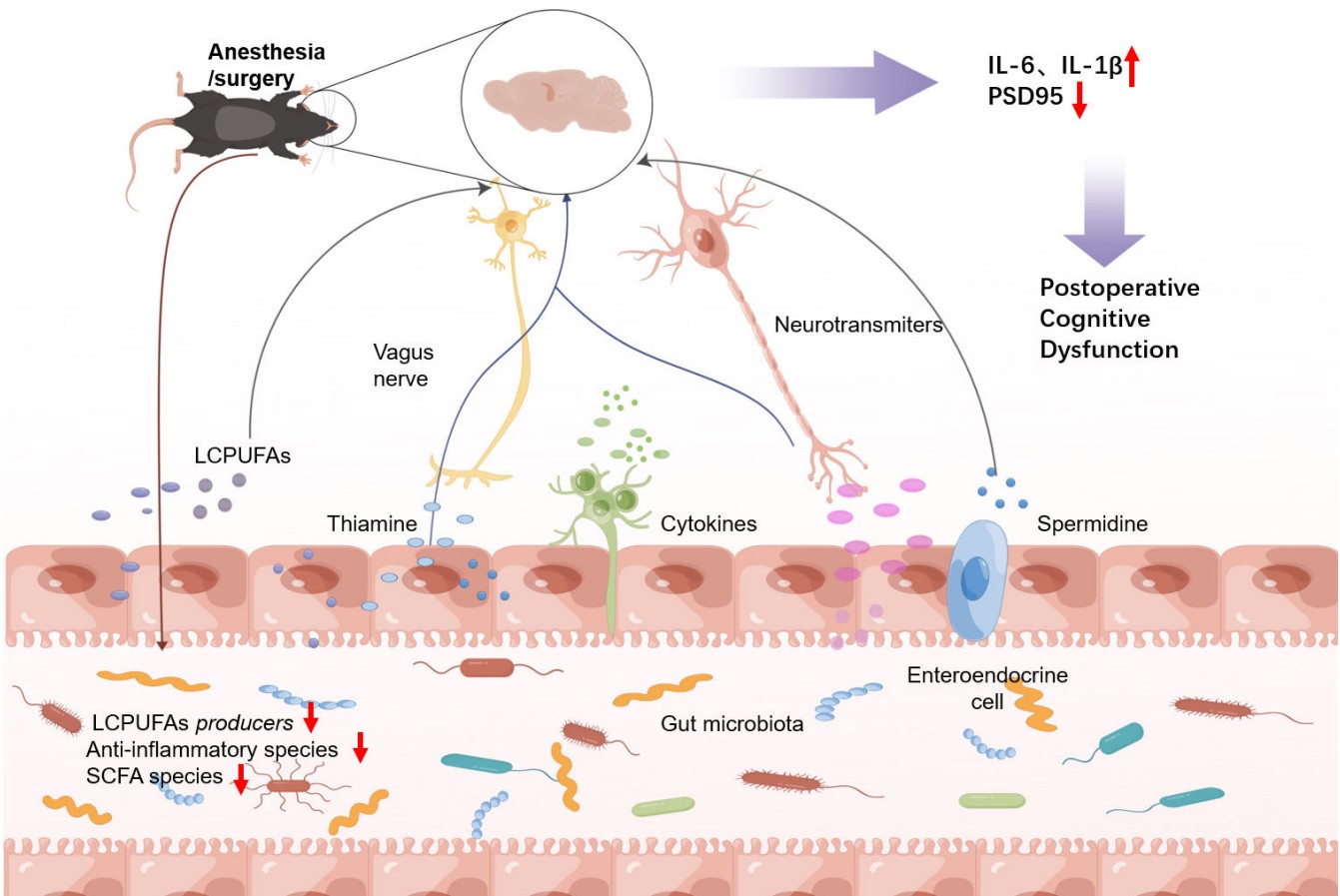

**FIG 11**  The potential underlying mechanism of postoperative cognitive impairment.

caused by an intestinal microbial imbalance. This work may contribute to identifying potential biomarkers for accurate diagnosis and treatment.

## MATERIALS AND METHODS

### Animals

The male C57BL/6 mice, aged 18 months, were obtained from Nanjing Medical University [SCXK(SU)2021–0001] and kept in a typical specific-pathogen-free animal facility under controlled environmental conditions (temperature: 22°C ± 2°C, humidity: 60% ± 5%, 12 hours light/dark cycle), with *ad libitum* access to food and water. To prevent the mixing of gut microbiota, each mouse was individually housed in its cage, and a control group and an experimental group were randomly assigned.

### Experimental protocols

The 18-month-old C57BL/6 mice were randomly divided into two groups: the control group ($n = 6$) received air treatment, and the experimental group ($n = 5$) underwent multiple AS procedures. The experimental mice were induced with 4% isoflurane and placed in an induction box until they had no corneal reflex. Subsequently, they were subjected to a combination of 1.5% isoflurane and oxygen administered through a nose cone for 2 hours. During this time, a laparotomy was performed to simulate surgical trauma commonly encountered in clinical settings. After suturing the wound, lidocaine was administered for pain relief. To maintain a body temperature of 37°C, a heat blanket was utilized. This continuous anesthetic method was repeated for 3 consecutive days. Upon waking up from anesthesia, the mice were allowed free access to food and water in their respective cages.

### Morris water maze tests

Spatial learning and memory assessment in mice was conducted using the MWMT. The test was conducted in a white plastic circular pool measuring 120 cm in diameter, 50 cm in height, and 22 cm deep, maintained at a constant temperature of 20°C. In the target quadrant, a transparent, circular platform with a diameter of 10 cm was placed approximately 0.5–1.0 cm below the water surface. Over 5 days, the mice underwent four trials per day, each in a randomly selected quadrant. To locate the hidden platform, the mice were allowed to swim for a maximum of 2 minutes during each trial. Once they had successfully found the platform, they were permitted to rest on it for 30 seconds. If the mice failed to find the platform within 30 seconds, they were returned to their respective cages. To analyze the time spent in the target quadrant and escape latency as indicators of reference memory, on day 6, we conducted a 2-minute probe test and recorded the hidden platform experiment. Using a free video recording system, we monitored the swimming paths of the mice and calculated the escape latency based on how long it took them to reach the platform.

### Open field test

To assess the exploratory activity and anxiety level of elderly mice, open-field tests are commonly used. In this test, the mice explored a white plastic chamber for 10 minutes positioned in the center (40 × 40 × 40 cm). Then, their exploratory behavior was monitored using a video tracking system (Shanghai Yuyan Instruments Co., Ltd., Shanghai, China). Following each test, the arena was disinfected with 75% alcohol to eliminate possible olfactory cues.

## Fecal sample collection

The samples were divided into two groups: the control group ($n$ = 6) and the multiple AS group ($n$ = 5). According to the clinical incidence of POCD and literature records, we chose to collect samples 72 hours after surgery (6). The specific operation is as follows: sterilize for 15 minutes, ventilate for 5 minutes, and alcohol wipe off the ultra-clean table. The abdominal massage method urges the animal to defecate on an ultra-clean bench, with at least five to six fecal samples. Put the samples into 1.5 mL sterile test tubes and quickly freeze them on dry ice. The samples should be stored at −80℃ within 2 hours after collection.

## DNA extraction and library construction

The hexadecyltrimethy ammonium bromide (CTAB) method was employed to extract microbial total DNA from 100 mg of mouse feces. Gel electrophoresis was used to evaluate the quality of the DNA extraction. Total DNA was eluted in 50 mL of the elution buffer and stored at −80℃. The PCR analysis was conducted by LC-BIO Technologies Co., Ltd. (Hangzhou, Zhejiang Province, China). The Qubit method (Hangzhou Lian Chuan Biotechnology Co., Ltd.) was employed to determine the DNA concentration, and the concentration was adjusted to 5 ng/μL. DNA libraries were constructed using the TruSeq Nano DNA LT Library Preparation Kit (FC-121–4001). The DNA has been fragmented through the use of double-stranded DNA (dsDNA) Fragmentase (NEB, M0348S), following an incubation of 30 minutes at 37℃. Subsequently, the fragments were randomly broken into sizes ranging from 200 to 500 bp. An "A" base was added to the 3' end, and a linker was added to the end of the DNA fragments after end-repair. To facilitate attachment to A-tailed DNA, each adapter had a T-base protruding portion. Adapters to complement single-end, paired-end, and indexed sequencing primer hybridization sites have been designed. The ligated products, consisting of single- or dual-index adapters, were then amplified using PCR. The amplification conditions were as follows: initial denaturation at 95℃ for 3 minutes; eight cycles of denaturation at 98°C for 15 seconds, annealing at 60℃ for 15 seconds, and extension at 72℃ for 30 seconds; and then a final extension at 72℃ for 5 minutes.

## Metagenomic analysis

The NovaSeq6000 platform (Illumina) was used for the high-throughput sequencing of the library after the QC process had been completed. To further analyze the reads obtained from the sequencing, the original sequence reads were filtered to obtain effective reads. The raw data were first split using Cutadapt (v1.9), followed by the removal of low-quality data with Fqtrim (v0.94). Host sequences were then eliminated using Bowtie2 (v2.2.0) to enhance the accuracy and precision of the species and functional annotation results. After data pre-processing, *de novo* assembly was performed on a single sample using Megahit (v1.2.9), and the assembled contigs were utilized for coding sequence prediction with MetaGeneMark (v3.26). Subsequently, clustering and de-duplication were carried out using CD-HIT (v4.6.1) based on the predicted results, resulting in a non-redundant UniGene set. Species annotation information was obtained by comparing the UniGene data set with the NR_mate library. Similarly, functional annotations of individual genes were obtained, encompassing a range of databases and resources. The Kyoto Encyclopedia of Genes and Genomes (KEGG-release 87.7) was used to annotate pathways, and the Carbohydrate-Active Enzymes (CAZy-2022.0806) database was used to analyze enzymes related to carbohydrates. The abundance spectra of single-feature genes, their classification, and their functional annotation are based on the classification and functional annotation of these single-feature genes. Fisher's exact test (non-replicated group and replicated group) was used to analyze the differences at each classification, functional, or gene level. QIIME2 was used to calculate alpha and beta diversity, and R packages were used to visualize the data. Species that differed significantly from one another were further compared using LEfSe analysis.

## Extraction of metabolites

By thawing the samples on ice and extracting them with a methanol buffer of 50%, the metabolites were extracted. An equal volume of the buffer was added to each sample, vortexed for a minute, then incubated at room temperature for 10 minutes to extract the metabolites. Protein precipitation was achieved by storing the mixture at −20°C overnight. Ten microliters of each sample were extracted, mixed into QC samples, and stored at −80°C until they were used in the ultra performance liquid chromatography/tandem mass spectrometry (UPLC-MS/MS) analysis.

## UPLC-MS/MS analysis

LC-MS systems were used to collect each sample sequentially. For all chromatographic separations, a Thermo Scientific high-performance liquid chromatography system was initially used. The reverse-phase separation was performed using an ACQUITY UPLC BEH C18 chromatographic column (100 × 2.1 mm, 1.8 mm; Waters, UK). With a flow rate of 0.4 mL/min, the column temperature was maintained at 35°C. The mobile phase consisted of solvent A (water with 0.1% formic acid) and solvent B (acetonitrile with 0.1% formic acid), which were used for elution. The gradient elution conditions were set as follows: 0–0.5 minutes, 5% B; 0.5–7 minutes, 5%–100% B; 7–8 minutes, 100% B; 8–8.1 minutes, 100%–5% B; 8.1–10 minutes, 5% B.

High-resolution tandem mass spectrometers (Q-Exactive, Thermo Scientific) were used to detect metabolites eluted from the chromatographic column. The Q-Exactive instrument has the capability to function in both positive and negative ion modes. In order to maintain the stability of the LC-MS system throughout the entire sample collection process, a QC sample, consisting of a pooled mixture of all samples, was collected after every 10 samples.

XCMS software was utilized to pre-process the obtained mass spectrometry data. This included peak picking, peak grouping, RT correction, secondary peak grouping, and isotope and adduct labeling. The LC-MS raw data files were converted to mzXML format and subsequently processed using the XCMS, CAMERA, and metaX toolboxes with R software (89). The RT and m/z data were combined to identify each ion. The intensity of each peak was measured and used to create a three-dimensional matrix that included peak indices (RT/m/z pairs), observation names (samples), and variables indicating ion intensity. The metabolites were identified through the online KEGG (https://www.kegg.jp/) and HMDB (http://www.hmdb.ca/) databases, with exact molecular mass data (m/z) matching the database information. If the mass difference between the observed and database values is below 10 ppm, the metabolite will be annotated. The molecular formula of the metabolite will be identified and confirmed through isotopic distribution measurements. Metabolite identifications were additionally validated using an in-house library of metabolite fragment profiles. MetaX pre-processing of peak data intensities was conducted (90). Features detected in less than 50% of the QC samples or less than 80% of the biological samples were removed, and the remaining peaks with missing values were imputed using the k-Nearest Neighbor algorithm to further improve data quality. Using the pre-processed data set, principal component analysis (PCA) was used for outlier detection and batch effect evaluation. A QC-based robust LOESS signal correction was applied to the data, considering the injection order, to reduce signal intensity drift. Moreover, the relative SDs of metabolic characteristics were calculated for all QC samples, and those exceeding 30% were eliminated.

Student $t$-tests were performed to identify variations in metabolite concentrations between two phenotypes. The $P$ value was corrected for multiple tests through an false discovery rate (FDR) [Benjamini-Hochberg (BH)]. Supervised PLS-DA was executed utilizing metaX to differentiate the diverse variables across the groups. The VIP value was computed, and an important feature selection was made using a VIP cutoff value of 1.0.

## Western blot analysis

Following the last inhalation of isoflurane for 72 hours, the mice were perfused with cold 0.9% normal saline. The hippocampus and prefrontal cortex were then isolated from the whole brain. The bicinchoninic acid (BCA )method was employed for protein quantification immediately following the total protein extraction. A 10% PAGE rapid preparation kit was utilized for the electrophoretic separation of proteins. Following transfer printing and blocking, the primary antibody was incubated overnight at 4°C. The primary antibodies used for western blot analysis included GAPDH (Cat. No. ab181602; 1:20,000; Abcam, UK) and anti-PSD95 (1:1,000; molecular weight 95 kDa; Cell Signal Transduction, Danvers, MA).

## RNA isolation and qPCR

The extraction of total RNA from the hippocampus was performed using TRIzol following the perfusion of the heart with 0.9% saline. Subsequently, cDNA was synthesized with a reverse transcription kit, and relative gene expression was measured by qPCR using the dye method. A relative quantification method (ΔΔCt) was used to quantify the mRNA expression of each gene, normalized to GAPDH. Sangon Biotech (Shanghai, China) provided all the primers, and the sequences are listed below.

IL-6: 5′-ACTTCCATCCAGTTGCCTTCTTG-3′ and 5′-TGTGTAATTAAGCCTCCGACTTGTG-3′;
IL-1β: 5′ -TCGCAGCAGCACATCAACAAG 3′ and 5′ -TCCACGGGAAAGACACAGGTAG-3′;
GAPDH: 5′-GTATGACTCCACTCACGGCAAA-3′ and 5′-GGTCTCGCTCCTGGAAGATG-3′

## Statistical analyses

The statistical methods employed to evaluate disparities between species or sample groups included Student's $t$ tests and Mann-Whitney $U$ tests. The calculation of alpha diversity for microbiota analysis was performed using QIIME. The analysis of beta diversity was conducted using ANOSIM. The calculation of a weighted UniFrac distance matrix and subsequent visualizations were performed using PCA. Using the R software, a heatmap was created based on levels of classification and species abundances. The species difference was analyzed by LEfSe analysis and the Mann-Whitney $U$ test. Metabolomics analysis involved the implementation of BH corrections, which encompassed univariate analysis of fold-changes and $t$ tests to derive $q$-values. Additionally, multivariate statistical analysis of VIP values obtained through PLS-DA was employed to discern metabolic ions that exhibited differential expression. We used R (version 3.6.3) and GraphPad Prism (version 8.0) for statistical analyses and figure creation. Lastly, Spearman correlation analysis was used for correlation analysis. Statistical significance was determined by an adjusted $P < 0.05$, unless otherwise stated. Data in the study are presented as mean values ± SEM.

### ACKNOWLEDGMENTS

We express our sincere gratitude to the Affiliated Hospital of Jiaxing University for generously providing us with an experimental platform to conduct our research. Additionally, we extend our appreciation to Lian Chuan Biotechnology Co., Ltd. for their invaluable technical support and guidance throughout our study.

This study was supported by the Zhejiang Provincial Basic Public Welfare Research Program (LGF21H090017), the Key Discipline Established with the Zhejiang Provincial Traditional Chinese Medical Innovation Team (No. 2022–19), and the Key disciplines of medicine in Jiaxing (Anesthesia; 2023-ZC-001).

The research was designed and the article framework was established by Q.Z. and J.H.. S.Z. and X.J. made equal contributions to the study, primarily involving the establishment and experimental verification of a cognitive impairment model, data analysis, and editing of the initial draft. Q.W. conducted literature research, fecal collection, and behavioral tests. J.J., L.X., and L.Y. were responsible for the analysis methods, corrections, and reviews of the research. The final manuscript was read and approved by all the authors.

## AUTHOR AFFILIATIONS

[1]Department of Anesthesiology and Pain Medicine, The Affiliated Hospital of Jiaxing University, Jiaxing, Zhejiang, China

[2]College of Life Science and Medicine, Zhejiang Sci-Tech University, Hangzhou, Zhejiang, China

## AUTHOR ORCIDs

Jun-gang Han  http://orcid.org/0009-0005-5609-9833
Qing-he Zhou  http://orcid.org/0000-0002-0731-6302

## AUTHOR CONTRIBUTIONS

Shi-hua Zhang, Data curation, Formal analysis, Investigation, Visualization, Writing – original draft | Xiao-yu Jia, Data curation, Formal analysis, Investigation, Writing – original draft | Qing Wu, Investigation, Software | Jia Jin, Writing – review and editing | Long-sheng Xu, Writing – review and editing | Lei Yang, Writing – review and editing | Jun-gang Han, Methodology, Project administration, Writing – review and editing | Qing-he Zhou, Funding acquisition, Methodology, Project administration, Writing – review and editing

## ETHICS APPROVAL

All studies and protocols (JUMC2020-024) were approved by the Laboratory Animal Ethics Committee of Jiaxing University Medicine College.

## ADDITIONAL FILES

The following material is available online.

### Supplemental Material

**Supplemental material (Spectrum03104-23-s0001.pdf).** Fig. S1 to S3.

### Open Peer Review

**PEER REVIEW HISTORY (review-history.pdf).** An accounting of the reviewer comments and feedback.

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
