## [Reviewer comments · Microbiology Spectrum]

Microbiology Spectrum

The involvement of the gut microbiota in postoperative cognitive dysfunction based on integrated metagenomic and metabolomics analysis

Shihua Zhang, Xiaoyu Jia, Qing Wu, Jia Jin, Long sheng Xu, Lei Yang, Jungang Han, and Qinghe Zhou

Corresponding Author(s): Qinghe Zhou, The affiliated hospital of Jiaying University

Review Timeline:

Submission Date:	August 22, 2023
Editorial Decision:	September 12, 2023
Revision Received:	October 11, 2023
Accepted:	October 23, 2023

Editor: Diyan Li

Reviewer(s): The reviewers have opted to remain anonymous.

Transaction Report:

DOI: <https://doi.org/10.1128/spectrum.03104-23>

September 12, 2023

Prof. Qinghe Zhou
The affiliated hospital of Jiaxing University
Anaesthesiology and Pain Medicine
1882 Zhonghuan South Road, Jiaxing City, China
Jiaxing, Zhejiang 314001
China

Re: Spectrum03104-23 (The involvement of the gut microbiota in postoperative cognitive dysfunction based on integrated metagenomic and metabolomics analysis)

Dear Prof. Qinghe Zhou:

Link Not Available

Sincerely,

Diyang Li

Journals Department
Reviewer comments:

Reviewer #1 (Comments for the Author):

This work is meaningful, but some of the minor mistakes are need to be addressed.

1. The number of the samples and the detection methods employed in the study should be clarified.
2. Detailed description of the mNGS technical are suggested to be provide, such as: sample collection, DNA extraction, primer sequences used in PCR amplification, and relevant parameters of the sequencing platform.
3. The relationship between aging, gut microbiota, and metabolites should be discussed. The limitations of the sample size or the mice model in the study are also suggested to be talked over.

4. The grammar and spelling need to be polished.

Reviewer #2 (Comments for the Author):

This study provides a significant reference for understanding cognitive impairments and emotional abnormalities in elderly individuals following surgery. Through a multi-tiered approach, the study comprehensively analyzes the underlying biological mechanisms associated with these changes. In the elderly population, the decline in cognitive function and the emergence of depressive symptoms after surgery have become common concerns. However, the fundamental causes of this phenomenon remain elusive. In this study, the authors have adeptly employed a diverse array of experimental methodologies, ranging from behavioral testing to molecular analysis, as well as metabolomics. These perspectives collectively offer a comprehensive and multifaceted exploration, delving deeper into the cognitive impairments and depressive behaviors exhibited by aged mice post anesthesia/surgery. By correlating observations of cognitive alterations with gut microbiota and metabolites, the authors establish a link between cognitive changes and intestinal microbial metabolites in elderly mice subjected to anesthesia/surgery. This not only opens new avenues for further research but also holds promise in guiding future intervention strategies and therapeutic approaches.

Strengths:

The authors have adopted a multi-faceted approach, furnishing a diverse and rich dataset brimming with details. This dataset includes behavioral testing, genetic analysis, and metabolomics, collectively shedding light on the cognitive impairments and depressive behaviors manifested by aged mice post anesthesia/surgery. Furthermore, the correlation of cognitive changes with microbial metabolites provides a novel view of investigation.

Deficiencies:

Minor revisions:

In the abstract, emphasize the novelty and importance of the study and briefly describe the application advantages of mNGS technology and main findings. Additionally, provide quantitative indicators to support the significance of the results, such as p-values or significance levels after FDR correction.

In the introduction, include relevant literature that highlights the existing knowledge on the relationship between gut microbiota and metabolites, and introduce previous studies to help readers better understand the scientific contribution and research motivation of this study.

Incorporate additional figures or visualizations in the results section to present data more intuitively. Ensure that the titles and captions of figures are clear and concise, facilitating reader comprehension and interpretation of the results.

Carefully proofread and make necessary corrections to grammar, spelling, punctuation, and other language errors in the manuscript.

Reviewer #3 (Comments for the Author):

Dr. Zhang and colleagues have presented a comprehensive analysis integrating metagenomics and metabolomics to elucidate the role of the gut microbiota in postoperative cognitive dysfunction. This study explores the complex relationship between the gut microbial community and changes in cognitive function following surgical interventions, revealing the crucial significance of the gut microbiota in postoperative cognitive dysfunction.

Major:

1. In the Results section, the description of the relationship between neuroinflammation, synaptic injury, gut microbiota, and changes in metabolites is not sufficiently clear. It would be helpful to provide more clarity and detail in explaining these connections.

2. In the Methods section, it would be beneficial to explain the rationale for choosing to collect samples at 72 hours after anesthesia. This information would provide a better understanding of the experimental design and help readers interpret the results.

3. Furthermore, including the incidence rate of postoperative cognitive dysfunction in the article would be valuable. This additional information would provide a better context for the findings and help assess the clinical relevance of the study.

Minor:

1. In the abstract, emphasize the novelty and importance of the study, and briefly describe the application advantages of mNGS technology and main findings. Additionally, provide quantitative indicators to support the significance of the results, such as p-values or significance levels after FDR correction.

2. In the introduction, include relevant literature that highlights the existing knowledge on the relationship between gut microbiota and metabolites, and introduce previous studies to help readers better understand the scientific contribution and research

motivation of this study.

3. In the section of metabolomics analysis, provide detailed descriptions of the metabolomics platform, databases utilized, and the analytical process. When identifying significantly different metabolites, include structural identification results and provide supporting literature references to ensure the reliability and biological significance of the metabolites.

4. Enhance the description of differences in microbial composition in the results section. Further annotate significantly distinct microbial taxa and provide statistical results of their relative abundances, such as using histograms or box plots to illustrate differences among samples.

5. There appears to be a minor error on line 134. Please carefully revise the writing to ensure accuracy and coherence.

6. In the discussion section, incorporate relevant literature to further explore the potential mechanisms underlying the relationship between aging, gut microbiota, and metabolites. Propose future research directions and potential applications. Additionally, clearly specify the limitations of the study, such as limited sample size or constraints of the mice model.

7. Additionally, I recommend thoroughly checking the figure legends and data units to ensure consistency and accuracy throughout the manuscript. This will enhance the clarity and understanding of the results presented.

8. Carefully proofread and make necessary corrections to grammar, spelling, punctuation, and other language errors in the manuscript.

Reviewer #4 (Comments for the Author):

Well-written manuscript. Timely and important theme and area of study. Mention the 'n' of independent experiments or animals used in each figure legend wherever applicable and in the materials and methods. If 'n' is 3 or less the authors must add more experiments to increase the 'n' to at least 5 to 6.

Staff Comments:

Preparing Revision Guidelines

Please return the manuscript within 60 days; if you cannot complete the modification within this time period, please contact me. If you do not wish to modify the manuscript and prefer to submit it to another journal, please notify me of your decision immediately so that the manuscript may be formally withdrawn from consideration by Microbiology Spectrum.

Dear Prof. Dr. Diyan Li,

We are grateful for the letter you sent and the reviewers' remarks about our manuscript, "*The involvement of the gut microbiota in postoperative cognitive dysfunction based on integrated metagenomic and metabolomics analysis (03104-23)*", which was submitted to *Microbiology Spectrum*.

These comments are all valuable and very helpful for revising and improving our manuscript, as well as the important guiding significance to our research. We have studied the comments carefully and made the necessary revisions, which we hope are completed to the Editors' satisfaction. The editors' and reviewers' comments were responded point-by-point, and modified portions are marked in red in the revised manuscript. The main corrections in the manuscript and the responses to the editors' and reviewers' comments are as follows.

We are looking forward to hearing from you.

Sincerely,

Qing-he Zhou

zqh10980@zjxu.edu.cn

Department of Anesthesiology and Pain Medicine, the Affiliated Hospital of Jiaxing University

No.1882, Ring 2nd South Road, Jiaxing City, Zhejiang Province 314000, China

Reviewer Comments

Reviewer #1

1. The number of the samples and the detection methods employed in the study should be clarified.

Response: Thank you for your valuable and helpful comments. We apologize for our carelessness. Thank you for the reminder. We have added a note on sample size in the methods section of the main text. It is as follows: "The 18-month-old C57BL/6 mice were randomly divided into two groups: the control group (n = 6) received air treatment, and the experimental group (n = 5) underwent multiple anesthesia/surgery procedures". See page 18, line 457-459.

2. Detailed description of the mNGS technical are suggested to be provide, such as: sample collection, DNA extraction, primer sequences used in PCR amplification, and relevant parameters of the sequencing platform.

Response: Thank you very much for your valuable comments. Based on your comments, we have explained and illustrated the mNGS technology as follows, which mainly includes sample collection, DNA extraction, DNA library building, and data analysis. In order to ensure the accuracy and reliability of the sequencing data from the source, we have a rigorous and reliable sample testing and quality control process, from DNA extraction to sequencing, and we strictly control the quality of the samples at each step to ensure the authenticity and reliability of the sequencing data. Specific details are described below:

Sample collection: “The samples were divided into two groups: the control group (n = 6) and the multiple AS group (n = 5). According to the clinical incidence of postoperative cognitive dysfunction and literature records, we chose to collect samples 72 hours after surgery (6). The specific operation is as follows: Sterilize for 15 minutes, ventilate for 5 minutes, and alcohol wipe off the ultra-clean table. The abdominal massage method urges the animal to defecate on an ultra-clean bench, with at least five to six fecal samples. Put the samples into 1.5 mL sterile test tubes and quickly freeze them on dry ice. The samples should be stored at -80°C within two hours after collection”. See page 20, line 487-494.

DNA extractions: “DNA from different samples was extracted using CTAB according to the manufacturer's instructions. The reagent, designed to uncover DNA from trace amounts of sample, has been shown to be effective for the preparation of DNA from most bacteria. Sample blanks consisted of unused swabs processed through DNA extraction and tested to contain no DNA amplicons. The total DNA was eluted in 50 µl of Elution buffer by a modification of the procedure described by the manufacturer (QIAGEN) and stored at -80°C until measurement in the PCR by LC-BIO Technologies (Hangzhou) Co., Ltd., Hangzhou, Zhejiang Province, China”.

DNA Library Construction: “A DNA library was constructed using the TruSeq Nano DNA LT Library Preparation Kit (FC-121-4001). DNA was fragmented by dsDNA

Fragmentase (NEB, M0348S) by incubating at 37°C for 30 minutes. Library construction began with fragmented cDNA. Blunt-end DNA fragments were generated using a combination of fill-in reactions and exonuclease activity, and size selection was performed with the provided sample purification beads. An A-base was then added to the blunt ends of each strand, preparing them for ligation to the indexed adapters. Each adapter contained a T-base overhang for ligation of the adapter to the A-tailed fragmented DNA. These adapters contained the full complement of sequencing primer hybridization sites for single, paired-end, and indexed reads. Single or dual index adapters were ligated to the fragments and the ligated products were amplified with PCR using the following conditions: initial denaturation at 95°C for 3 minutes; 8 cycles of denaturation at 98°C for 15 seconds, annealing at 60°C for 15 seconds, and extension at 72°C for 30 seconds; and then a final extension at 72°C for 5 minutes”. See page 20-21, line 495-511.

Data analysis: “The NovaSeq6000 platform (Illumina) was used for the high-throughput sequencing of the library after the quality control process had been completed. To further analyze the reads obtained from the sequencing, the original sequence reads were filtered to obtain effective reads. The raw data were first split using Cutadapt (v1.9), followed by the removal of low-quality data with Fqtrim (v0.94). Host sequences were then eliminated using Bowtie2 (v2.2.0) to enhance the accuracy and precision of the species and functional annotation results. After data pre-processing, de novo assembly was performed on a single sample using Megahit (v1.2.9), and the assembled contigs were utilized for coding sequence (CDS) prediction with MetaGeneMark (v3.26). Subsequently, clustering and de-duplication were carried out using CD-HIT (v4.6.1) based on the predicted results, resulting in a non-redundant UniGene set. Species annotation information was obtained by comparing the UniGene dataset with the NR_mate library. Similarly, functional annotations of individual genes were obtained, encompassing a range of databases and resources. The Kyoto Encyclopedia of Genes and Genomes (KEGG-release 87.7) was used to annotate pathways, and the Carbohydrate-Active Enzymes (CAZy-2022.0806) database was used to analyze enzymes related to carbohydrates. The abundance spectra of single-feature genes, their classification, and their functional annotation are based on the

classification and functional annotation of these single-feature genes. Fisher's exact test (non-replicated group and replicated group) was used to analyze the differences at each classification, functional, or gene level. QIIME2 was used to calculate alpha and beta diversity, and R packages were used to visualize the data. Species that differed significantly from one another were further compared using linear discriminant analysis (LDA) effect size (LEfSe) analysis". See page 21-22, line 512-532.

3. The relationship between aging, gut microbiota, and metabolites should be discussed. The limitations of the sample size or the mice model in the study are also suggested to be talked over.

Response: We sincerely appreciate your valuable comments. We think it is a good suggestion. We have carefully reviewed the literature and added more information about the link between aging, gut flora and metabolites and provided references to support it in the discussion section of the revised manuscript. The details are as follows: "Ageing leads to a deterioration of cellular, tissue, and organ functions, accompanied by susceptibility to chronic inflammation in the central nervous system and gastrointestinal system, which are particularly sensitive to metabolic dysregulation. With ageing, there is an increase in the number of goblet cells, while the expression of alpha-defensins, lysozymes, and F4/80 mRNA, along with NOx levels and protein concentrations of tight junction proteins, decreases (31, 32). These changes are associated with elevated intestinal permeability and increased levels of bacterial endotoxins (33). Several studies have linked these age-related modifications to alterations in the gut microbiome observed in elderly humans and animals (34-36). In a recent study, it was found that transplanting fecal matter from aged individuals into young mice leads to disruption of the intestinal epithelial barrier and heightened levels of inflammation, particularly in the retina and nervous system. Neuroinflammation in older mice with young donor microbiota was reversed by enriching B vitamins and lipid synthesis pathways, implying an important role in gut microbial metabolism in aging (37). Additionally, postoperative cognitive dysfunction may possess multifactorial origins, such as an intraoperative inflammatory response, anesthesia, and aging, further increasing susceptibility. In this study, we aim to investigate the contribution of gut microbes to postoperative cognitive dysfunction in elderly patients". See page 12, line 275-290.

We also provide a full discussion of the limitations of this study. Specific details are given below: “There are some limitations to this study. On the one hand, small sample sizes may make some differences appear larger. On the other hand, there are certain differences between animal models and human diseases that do not fully simulate clinical outcomes”. See page 18, line 440-443.

4. The grammar and spelling need to be polished.

Response: Thank you for reviewer’s significant reminding. The manuscript has been carefully revised by a professional language editing service to improve the grammar and readability. These changes do not affect the content or framework of the paper. We do not list the changes here, but they are marked in red in the revised paper. We sincerely thank the reviewers for their enthusiastic work and hope that the revisions will meet with your approval. Thanks again for your valuable comments and suggestions!

Reviewer # 2

1. In the abstract, emphasize the novelty and importance of the study and briefly describe the application advantages of mNGS technology and main findings. Additionally, provide quantitative indicators to support the significance of the results, such as p-values or significance levels after FDR correction.

Response: We sincerely appreciate your valuable comments and suggestions. In this study, we used macro-genomic techniques to characterize the changes and differences in the intestinal flora of postoperative cognitively dysfunctional aged mice. We characterized the differential flora to species-level differences and demonstrated their functional differences as well, which was not done in previous studies. The specific description of the macro-genomic technology is shown in detail in our Methods, which mainly includes sample collection, DNA extraction, DNA library creation, and macro-genomic sequencing analysis. See page 20-22, lines 487-532 for more information. For the analysis of the differential colonies, we used different methods, mainly including Stamp analysis based on $P < 0.05$ and $\log_2FC > 1$ and LEfSE analysis based on $P < 0.05$ and $LDA > 3$ to fully demonstrate the colonies with significant differences. The significance of differences of all data in the study was demonstrated on the basis of $P < 0.05$. Differences such as behavioral statistical analysis, Western blot statistical analysis, and statistical analysis of PCR results were expressed as follows: *, $P < 0.05$; **, $P < 0.01$; ***, $P < 0.001$; ****, $P < 0.0001$. We will do our best to fully demonstrate the results of our study. Thank you for your careful reading and valuable comments.

2. In the introduction, include relevant literature that highlights the existing knowledge on the relationship between gut microbiota and metabolites, and introduce previous studies

to help readers better understand the scientific contribution and research motivation of this study.

Response: We appreciate your professional comments on our articles, which are used to improve the quality of the manuscript. We have extensively revised the manuscript based on your suggestions, i.e., we have included a preface about the link between gut flora and metabolites. And references have been introduced to support the arguments. The details are as follows: “Research into the brain-gut axis, which investigates the interaction between the brain and the gut, has gained significance in recent years. Humans coexist in symbiosis with diverse microorganisms residing in the intestines, including bacteria, yeasts, archaea, viruses, protozoa, and even parasites such as worms, collectively referred to as the intestinal microbiota (13, 14). They have a reciprocal relationship with the host, with over 10^{14} bacterial cells distributed throughout the gastrointestinal tract (GI), with the vast majority (10^{10} - 10^{12} CFU/g of intestinal contents) located in the ileum and colon (15). The gut microbiota transforms various dietary components, such as macronutrients, micronutrients, fiber, and polyphenols, into a multitude of metabolites, including amino acid derivatives, vitamins, short-chain fatty acids, and trimethylamine (16). These metabolites and dietary components, derived from microbes, play critical roles in regulating the host's homeostasis, particularly with respect to the blood-brain barrier integrity and brain function (17–19). When the microbial balance is disturbed, the body's physiological metabolism undergoes corresponding changes, which are then communicated to the brain via circulatory or neural pathways. Therefore, there exists a co-metabolic process whereby the body's metabolism is influenced by both its own functioning as well as the presence of the intestinal flora”. See page 4, line 79-93.

3. Incorporate additional figures or visualizations in the results section to present data more intuitively. Ensure that the titles and captions of figures are clear and concise, facilitating reader comprehension and interpretation of the results.

Response: Thank you for your valuable and helpful comments. Thank you for the reminder. We've shown all the results in visual images in the article. It makes all the data more visual and intuitive. The captions and legends of the graphs are clear and concise.

Thanks again for your suggestion.

4. Carefully proofread and make necessary corrections to grammar, spelling, punctuation, and other language errors in the manuscript.

Response: We are very grateful for the reviewer's careful reading. We have made careful modifications to the original manuscript and carefully proofread it to minimize typographical and grammatical errors. We have asked native English speakers to polish and modify the manuscript, and they are marked in red in the revised paper. We sincerely thank the reviewers for their enthusiastic work and hope that the revisions will meet with your approval. Thanks again for your valuable comments and suggestions!

Reviewer # 3

1. In the Results section, the description of the relationship between neuroinflammation, synaptic injury, gut microbiota, and changes in metabolites is not sufficiently clear. It would be helpful to provide more clarity and detail in explaining these connections.
Response: We sincerely appreciate your careful reading of our manuscript. Regarding your statement about the relationship between gut flora, metabolites, and neuroinflammation, we have addressed it in the Discussion and provided references to support it. Specifically, they are as follows: “Cognitive dysfunction usually occurs after surgery in elderly patients, characterized by impaired memory, logical thinking, hallucinations, and delusions (38). In this study, we examined the behavior, intestinal flora, metabolites, IL-6 and IL-1 β hippocampal levels, and synaptic function of elderly mice with anesthesia/surgery-induced cognitive dysfunction. Our findings revealed significant behavioral alterations, neuroinflammation, synaptic dysfunction, and disruptions in the intestinal flora and metabolites in the aged mice compared to normal mice. Specifically, older mice exhibited a longer escape latency and a shorter time spent in the target quadrant in the Morris water maze test. This finding is consistent with previous studies (39). Moreover, synaptic damage was observed in the hippocampus and the prefrontal cortex, further supporting the presence of cognitive impairments. Elevated levels of IL-6 and IL-1 β were also detected in the hippocampus, suggesting the occurrence of neuroinflammation. Neuroinflammation has been associated with memory impairment, brain injury, depression, and other neuropsychiatric disorders (40-45). Studies have proposed that changes in the intestinal flora can induce neuroinflammation, leading to the release of pro-inflammatory cytokines including CRP, IL-1 β , IL-6, and TNF- α , and the disruption of the blood-brain barrier, which may activate or impair astrocytes and contribute to neurodegenerative disorders (46, 47). These findings suggest a potential link between surgery/anesthesia-induced postoperative cognitive dysfunction and alterations in the gut flora. However, further research is necessary to fully elucidate the underlying mechanisms”. See page 12-13, line 291-308.

2. In the Methods section, it would be beneficial to explain the rationale for choosing to collect samples at 72 hours after anesthesia. This information would provide a better understanding of the experimental design and help readers interpret the results.

Response: Your suggestions are much appreciated. We have introduced references in the Methods section of the text to support that sampling and behavioral validation at 72 hours postoperatively is beneficial. Postoperative cognitive dysfunction is one of the most common neurological complications after anesthesia in older adults. Clinically, it has been found that in elderly people who have undergone major surgeries such as cardiac surgery, hip replacement and some other orthopedic surgeries, they may experience depressed mood, faulty memory and inability to concentrate. This phenomenon is extremely obvious in the first week after surgery. In animals, many studies have found that cognitive deficits are most pronounced in older mice at 3 days after surgery. The difference in incidence between humans and animals may be due to genetic variability between species. Therefore, in our study we also chose to perform behavioral tests at 72 hours postoperatively to assess their postoperative cognitive abilities.

3. Furthermore, including the incidence rate of postoperative cognitive dysfunction in the article would be valuable. This additional information would provide a better context for the findings and help assess the clinical relevance of the study.

Response: Thank you very much for your suggestions and comments. Based on your suggestions we have included a relevant discussion about the clinical incidence of postoperative cognitive dysfunction in the introduction. This will enhance the research significance and improve the quality of our article. Thank you again for your constructive comments. The details are as follows: "Clinical evidence indicates that Postoperative Cognitive Dysfunction affects 10-60% of elderly patients within the first week following surgery, with one-third experiencing long-term dysfunction (5, 6). Therefore, understanding the molecular mechanisms of postoperative cognitive dysfunction is crucial for the search for potential therapeutic targets and clinical interventions". See page 3, line 66-70".

4. In the abstract, emphasize the novelty and importance of the study, and briefly describe the application advantages of mNGS technology and main findings. Additionally, provide quantitative indicators to support the significance of the results, such as p-values or significance levels after FDR correction.

Response: We sincerely appreciate your valuable comments and suggestions. In this study, we used macro-genomic techniques to characterize the changes and differences in the intestinal flora of postoperative cognitively dysfunctional aged mice. We characterized the differential flora to species-level differences and demonstrated their

functional differences as well, which was not done in previous studies. The specific description of the macro-genomic technology is shown in detail in our Methods, which mainly includes sample collection, DNA extraction, DNA library creation, and macro-genomic sequencing analysis. See page 20-22, lines 487-532 for more information. For the analysis of the differential colonies, we used different methods, mainly including Stamp analysis based on $P < 0.05$ and $\log_2FC > 1$ and LEfSE analysis based on $P < 0.05$ and $LDA > 3$ to fully demonstrate the colonies with significant differences. The significance of differences of all data in the study was demonstrated on the basis of $P < 0.05$. Differences such as behavioral statistical analysis, Western blot statistical analysis, and statistical analysis of PCR results were expressed as follows: *, $P < 0.05$; **, $P < 0.01$; ***, $P < 0.001$; ****, $P < 0.0001$. We will do our best to fully demonstrate the results of our study. Thank you for your careful reading and valuable comments.

5. In the introduction, include relevant literature that highlights the existing knowledge on the relationship between gut microbiota and metabolites, and introduce previous studies to help readers better understand the scientific contribution and research motivation of this study.

Response: We appreciate your professional comments on our articles, which are used to improve the quality of the manuscript. We have extensively revised the manuscript based on your suggestions, i.e., we have included a preface about the link between gut flora and metabolites. And references have been introduced to support the arguments. The details are as follows: "Research into the brain-gut axis, which investigates the interaction between the brain and the gut, has gained significance in recent years. Humans coexist in symbiosis with diverse microorganisms residing in the intestines, including bacteria, yeasts, archaea, viruses, protozoa, and even parasites such as worms, collectively referred to as the intestinal microbiota (13, 14). They have a reciprocal relationship with the host, with over 10^{14} bacterial cells distributed throughout the gastrointestinal tract (GI), with the vast majority (10^{10} - 10^{12} CFU/g of intestinal contents) located in the ileum and colon (15). The gut microbiota transforms various dietary components, such as macronutrients, micronutrients, fiber, and polyphenols, into a multitude of metabolites, including amino acid derivatives, vitamins, short-chain fatty acids, and trimethylamine (16). These metabolites and dietary components, derived from microbes, play critical roles in regulating the host's homeostasis, particularly with respect to the blood-brain barrier integrity and brain function (17–19). When the microbial balance is disturbed, the body's physiological metabolism undergoes corresponding changes, which are then communicated to the brain via circulatory or neural pathways. Therefore, there exists a co-metabolic process whereby the body's metabolism is influenced by both its own functioning as well as the presence of the intestinal flora". See page 4, line 79-93.

6. In the section of metabolomics analysis, provide detailed descriptions of the metabolomics platform, databases utilized, and the analytical process. When identifying significantly different metabolites, include structural identification results and provide supporting literature references to ensure the reliability and biological significance of the metabolites.

Response: Thank you very much for your input. We apologize for the lack of clarity in our presentation of the metabolome in the Methods section. We have provided a detailed description of your proposed metabolomics below:

LC-MS systems were used to collect each sample sequentially. For all chromatographic separations, a Thermo Scientific high-performance liquid chromatography system was initially used. The reverse-phase separation was performed using an ACQUITY UPLC BEH C18 chromatographic column (100 mm x 2.1 mm, 1.8 mm; Waters, UK). With a flow rate of 0.4 mL/min, the column temperature was maintained at 35 °C. The mobile phase consisted of solvent A (water with 0.1% formic acid) and solvent B (acetonitrile with 0.1% formic acid), which were used for elution. The gradient elution conditions were set as follows: 0–0.5 min, 5% B; 0.5–7 min, 5%–100% B; 7–8 min, 100% B; 8–8.1 min, 100%–5% B; 8.1–10 min, 5% B.

High-resolution tandem mass spectrometers (Q-Exactive, Thermo Scientific) were used to detect metabolites eluted from the chromatographic column. The Q-Exactive instrument has the capability to function in both positive and negative ion modes. In order to maintain the stability of the LC-MS system throughout the entire sample collection process, a quality control (QC) sample, consisting of a pooled mixture of all samples, was collected after every 10 samples.

XCMS software was utilized to pre-process the obtained mass spectrometry data. This included peak picking, peak grouping, retention time correction, secondary peak grouping, and isotope and adduct labeling. The LC-MS raw data files were converted to mzXML format and subsequently processed using the XCMS, CAMERA, and metaX toolboxes with R software (89). The retention time (RT) and m/z data were combined to identify each ion. The intensity of each peak was measured and used to create a three-dimensional matrix that included peak indices (retention time/m/z pairs), observation names (samples), and variables indicating ion intensity. The metabolites were identified through the online KEGG (<https://www.kegg.jp/>) and HMDB (<http://www.hmdb.ca/>) databases, with exact molecular mass data (m/z) matching the database information. If the mass difference between the observed and database values is below 10 ppm, the metabolite will be annotated. The molecular formula of the metabolite will be identified and confirmed through isotopic distribution measurements. Metabolite identifications were additionally validated using an in-house library of metabolite fragment profiles. MetaX pre-processing of peak data intensities was conducted (90). Features detected in

less than 50% of the QC samples or less than 80% of the biological samples were removed, and the remaining peaks with missing values were imputed using the k-Nearest Neighbor algorithm to further improve data quality. Using the pre-processed data set, principal component analysis (PCA) was used for outlier detection and batch effect evaluation. A QC-based robust LOESS signal correction was applied to the data, considering the injection order, to reduce signal intensity drift. Moreover, the relative standard deviations of metabolic characteristics were calculated for all QC samples, and those exceeding 30% were eliminated.

Student t-tests were performed to identify variations in metabolite concentrations between two phenotypes. The P value was corrected for multiple tests through an FDR (Benjamini-Hochberg). Supervised PLS-DA was executed utilizing metaX to differentiate the diverse variables across the groups. The VIP value was computed, and an important feature selection was made using a VIP cutoff value of 1.0. See page 22-24, line 540-578.

We have placed the structures of the identified metabolites in a miscellaneous file.

7. Enhance the description of differences in microbial composition in the results section. Further annotate significantly distinct microbial taxa and provide statistical results of their relative abundances, such as using histograms or box plots to illustrate differences among samples.

Response: We sincerely thank you for your suggestions and comments. Regarding the description of the differential flora, we used different statistical methods to demonstrate in the article, including a Stamp analysis and an LEfSe analysis for the differential flora. Stamp analysis was based on differential flora screened by $P < 0.05$ and $\log_2FC > 1$, which was mainly displayed in Figure 4. While LEfSe analysis was based on differential flora screened by $P < 0.05$ and $LDA > 3$, and the results were displayed in Figure 5. The results obtained from the two analysis methods were basically the same. We also provide box plots of the relative abundance shown in the Supplementary Material (Figure S1).

8. There appears to be a minor error on line 134. Please carefully revise the writing to ensure accuracy and coherence.

Response: Thank you for your careful reading and review, and we apologize profusely for our carelessness. We appreciate your careful review of our work. We have made revisions to enhance the academic quality of our text. Thank you for your feedback.

9. In the discussion section, incorporate relevant literature to further explore the potential mechanisms underlying the relationship between aging, gut microbiota, and metabolites. Propose future research directions and potential applications. Additionally, clearly specify the limitations of the study, such as limited sample size or constraints of the mice model.

Response: We sincerely appreciate your valuable comments. We have added to the discussion about the relationship between aging, gut flora and metabolism. On the other

hand, we have also discussed and reflected on the limitations of our study as well. The details are as follows: “Ageing leads to a deterioration of cellular, tissue, and organ functions, accompanied by susceptibility to chronic inflammation in the central nervous system and gastrointestinal system, which are particularly sensitive to metabolic dysregulation. With ageing, there is an increase in the number of goblet cells, while the expression of alpha-defensins, lysozymes, and F4/80 mRNA, along with NOx levels and protein concentrations of tight junction proteins, decreases (31, 32). These changes are associated with elevated intestinal permeability and increased levels of bacterial endotoxins (33). Several studies have linked these age-related modifications to alterations in the gut microbiome observed in elderly humans and animals (34-36). In a recent study, it was found that transplanting fecal matter from aged individuals into young mice leads to disruption of the intestinal epithelial barrier and heightened levels of inflammation, particularly in the retina and nervous system. Neuroinflammation in older mice with young donor microbiota was reversed by enriching B vitamins and lipid synthesis pathways, implying an important role in gut microbial metabolism in aging (37). Additionally, postoperative cognitive dysfunction may possess multifactorial origins, such as an intraoperative inflammatory response, anesthesia, and aging, further increasing susceptibility. In this study, we aim to investigate the contribution of gut microbes to postoperative cognitive dysfunction in elderly patients”. See page 12, line 275-290. The limitations of the study as a whole are also discussed. It is specified below: “There are some limitations to this study. On the one hand, small sample sizes may make some differences appear larger. On the other hand, there are certain differences between animal models and human diseases that do not fully simulate clinical outcomes”. See page 18, line 440-443.

10. Additionally, I recommend thoroughly checking the figure legends and data units to ensure consistency and accuracy throughout the manuscript. This will enhance the clarity and understanding of the results presented.

Response: Thank you very much for your advice. We have thoroughly checked the diagrams and legends throughout the article so that the manuscript corresponds to the images.

11. Carefully proofread and make necessary corrections to grammar, spelling, punctuation, and other language errors in the manuscript.

Response: Thank you for your suggestions and careful reading. In order to improve the quality of the manuscript, we invited a team of professional embellishers to grammatically check and revise the manuscript. We did our best to improve the manuscript and made some changes. These changes do not affect the content or framework of the paper. We do not list these changes here, but they are marked in red in the revised paper. We sincerely thank the reviewers for their enthusiastic work and hope

that the revised manuscript will be recognized by you. Thank you again for your valuable comments and suggestions!

Reviewer # 4

1. Well-written manuscript. Timely and important theme and area of study. Mention the 'n' of independent experiments or animals used in each figure legend wherever applicable and in the materials and methods. If 'n' is 3 or less the authors must add more experiments to increase the 'n' to at least 5 to 6.

Response: Thank you very much for your valuable comments and we apologize for our carelessness. We have labeled "n = 6" in the Figure Notes and Methods section of the article. Please see the revised manuscript for details. Thank you again for your careful reading and valuable comments, which are crucial for improving the quality of the article.

Re: Spectrum03104-23R1 (The involvement of the gut microbiota in postoperative cognitive dysfunction based on integrated metagenomic and metabolomics analysis)

Dear Prof. Qinghe Zhou:

Your manuscript has been accepted, and I am forwarding it to the ASM production staff for publication. Your paper will first be checked to make sure all elements meet the technical requirements. ASM staff will contact you if anything needs to be revised before copyediting and production can begin. Otherwise, you will be notified when your proofs are ready to be viewed.

Sincerely,
Diyan Li
Editor
Microbiology Spectrum

Reviewer #2 (Comments for the Author):

I have checked the authors' responses to my previous suggestions, and I am content with their precise and targeted responses. The authors provided a comprehensive explanation of the reliability and effectiveness of these methods and collection techniques. The authors have also verified and enhanced the accuracy and completeness of the citations. Regarding language usage, the authors have enhanced the revised draft. The language is now standardized, enhancing the article's readability and comprehensibility.

Based on these notable improvements, I believe that this revised manuscript aligns with the standards of our journal. I recommend accepting the revised draft for publication.

Reviewer #3 (Comments for the Author):

I have received and reviewed the authors' responses to the issues and suggestions I previously raised. I believe they have adequately addressed all the key points of concern. The revised manuscript shows improvement in exposition and technical

details, particularly with their more comprehensive explanations on the choice of research methods, the rationale behind the experimental design, and the statistical significance of the results.

Moreover, they have made important revisions to the structure and language quality of the paper, which have clarified the presentation of the research content and findings. I especially appreciate the inclusion of more relevant information in the discussion section to further explore the potential mechanisms underlying the relationship between aging, gut microbiota, and metabolites, providing a deeper context for the scientific contribution of this study.

In light of these improvements, I am convinced that the manuscript now meets the publication standards of this journal.